# Number Value Loss in LLMs and N-adic Tokenization

## Abstract

This paper provides a theoretical analysis of metrics for numerical value comparison and reasoning in machine learning. We address the root causes of optimization failure in numerical representations through three primary contributions.

First, we introduce MALL (Magnitude-Aware Log Loss), a metric designed to ensure gradient stability and sensitivity across more than 30 orders of magnitude. We demonstrate that MALL maintains a robust signal for both global magnitude and local precision across the entire $\mathbb{R}^+$ domain, resolving the vanishing and exploding gradient problems inherent in traditional metrics. This ensures a stable foundation for numerical reasoning of bijective to number value tokenizations, decoded token sequences or regressions and makes MALL a versatile tool for numerical comparison, applicable both as an auxiliary component within other metrics and as a standalone baseline.

Second, we identify the Softmax boundary problem — a fundamental structural failure at digit-order transitions caused by the interplay between independent positional distributions and positional tokenization. We establish a No-Go theorem proving that additive per-token continuous losses are mathematically incompatible with numerical stability over large ranges. Consequently, we demonstrate that structured discontinuities in the gradient field act as a necessary catalyst for global consistency and propose a deferred global loss with hardmax as a regularization strategy to stabilize this behavior.

Third, we propose a geometrical embedding regularizer, Triangle Loss, based on the triangle inequality to enforce numerical continuity within the embedding manifold. By ensuring that the geometric relationships between embeddings reflect their numerical distances, Triangle Loss improves generalization for rare tokens in any numerical bijective tokenization and provides a structural basis for learning numerical proximity at extreme scales.

Through mathematical proofs and gradient field visualizations, we demonstrate that our framework addresses the fundamental limitations of current numerical objectives, providing a robust foundation for coherent numerical intelligence in neural architectures.

**Keywords:** LLM Regression, MALL (Magnitude-Aware Log Loss), Triangle Loss, Softmax Boundary Problem, Numerical Reasoning, Scale-Invariance, Numerical Tokenization.

## 1 Introduction

Standard subword tokenization methods—such as Byte Pair Encoding (BPE)—fragment numerical values arbitrarily, disrupting their inherent decimal structure (Sennrich et al., 2015; Bhatia et al., 2025). This lack of structural awareness in state-of-the-art models like Llama 3 (Dubey et al., 2024) forces neural architectures to learn complex, non-positional mappings, leading to significant errors in numerical reasoning tasks (Zhang et al., 2025). Systematic empirical probing reveals a stark "scale-blindness" in Large Language Models (LLMs): while architectures maintain proficiency on short, low-digit numbers, their arithmetic accuracy decays rapidly as the operand length scales up, rendering models unable to handle multi-digit arithmetic. Furthermore, shifting numerical scales outside the standard training distribution triggers up to a 14 percentage point increase in pure logical reasoning errors, exposing severe optimization and representational

vulnerabilities at both extreme macro-scales and tiny micro-scales (Shrestha et al., 2025). While N-adic Suffix Tokenization (NST) addresses this representation barrier by providing a deterministic, bijective mapping between compound tokens and their numerical values, the structural fix is only half of the solution (Chetverina, 2026). Because the model still operates under traditional objective functions, it remains fundamentally scale-blind and unable to maintain uniform gradient sensitivity across extreme magnitude variations.

We argue that one of the primary reasons for this systemic failure is fundamentally rooted in the optimization objective. We show that even with structurally sound tokenization or continuous representations like xVal (Izacard et al., 2023), traditional loss functions analytically fail to transmit coherent error signals across diverse scales. Because the loss function dictates the entire gradient surface during backpropagation, any mathematical deficiency in its objective directly compromises the optimization trajectory. Specifically, standard objectives generate a highly volatile or vanishing gradient field across large magnitude spectra, making it mathematically impossible for the network weights to converge uniformly. Therefore, a robust numerical intelligence cannot be achieved by architectural adjustments or data scaling alone; it strictly requires specialized objectives that maintain a stable, scale-invariant gradient signal across astronomical ranges.

## 1.1 Motivation: Why Existing Metrics Fail

Our investigation into multi-scale numerical learning revealed three fundamental bottlenecks that existing frameworks fail to address:

- *Token-level vs. Value-level Failure:* Initially, our research aimed to leverage established numerical loss functions, specifically NTL (Numerical Token Loss)(Zausinger et al., 2025), which operates directly on tokens, within the context of N-adic representations. However, empirical and analytical probing revealed critical deficiencies: in vast numerical ranges, a linear distance either leads to gradient explosion or becomes negligible. The same problem arises with robust losses, such as the Huber loss (Huber, 1964) or its differentiable analogue log-cosh (Chen et al., 2019). Moving to a higher abstraction, we considered xVal tokenization of numbers (Izacard et al., 2023), which employs MSE with linear value prescaling. We also considered some of the metrics from a review on loss functions for ML (Li et al., 2025). While certain logarithmic variants capture scale information, they are not sensitive to local differences, especially for tiny numbers.

- *The MALL Solution:* To transcend these limitations, we developed MALL (Magnitude-Aware Log Loss). By integrating a dynamic triadic denominator and a cubic power-law penalty, MALL ensures that the gradient remains informative across at least the entire 33-order spectrum of $\mathbb{R}^+$.

- *The Softmax Boundary Problem:* During stress-testing, we observed some numerical instabilities at digit-level transitions when a sum of loss is calculated for the whole number consisting of several tokens. We provide a formal No-Go theorem proving that no additive continuous loss can resolve this without triggering gradient instability. However, we further show that this feature — in the form of penalties for such digit-level transitions — actually allows us to control the errors that Softmax would otherwise generate due to the independence of the final token probabilities that constitute a single number. But for a slight regularization strategy we propose a Deferred Numerical Loss (DNL).

- *Numerical Continuity in Embedding Space:* Finally, for tokenization schemes that are bijective to real numerical values (such as N-adic Suffix Tokenization), we argue that the model should leverage the inherent continuity of neural architectures. Since a neural network itself implements a continuous function from the embedding space to its output, any discontinuity or non-locality in the embedding mapping directly undermines the network's ability to generalize across numerical scales. Formally, the mapping from a numerical value to its token embedding must be continuous: numerically close values must map to proximal points in the embedding space. This requirement directly addresses the documented failure of standard embeddings to capture numerical scales (Wallace et al., 2019). To enforce this property, we propose Triangle Loss — a direct regularization of the embedding manifold. Beyond general stability, its primary objective is to alleviate the "rare token" problem, which is particularly prevalent at extreme magnitudes in N-adic suffix tokenization. Geometrical

anchoring of rare tokens between their numerical neighbors ensures that the model generalizes across the entire numerical axis, even in the absence of frequent observations for specific magnitudes.

## 2 Preliminaries: N-adic Suffix Tokenization

In this section, we define the tokenization framework used as a representative case to analyze the relationship between discrete tokens and continuous values. We adopt the N-adic Suffix Tokenization (NST) scheme (Chetverina, 2026) as a formal basis, as its bijective, per-token structure is essential for the derivation of several proofs presented in this paper.

A critical property of the NST framework is that each token maps to a unique and fixed numerical value determined by its suffix. Unlike standard BPE-based numerical representations, where a digit's value depends on its relative shift from a decimal point within a larger string, each NST token is a self-contained numerical entity. This allows us to treat every token as an independent number with a predefined magnitude, rather than a constituent fragment of an ambiguous sequence.

While NST provides the necessary formal grounding for our theoretical analysis, the proposed MALL metric is designed to be more general. As such, it functions as a robust metric for comparing numerical values across the $\mathbb{R}^+$ domain.

To explore the effects of scale and granularity, we focus on two specific cases of Option B :

- *Triadic (N = 3), TST* A "human-like" variant that serves as the primary focus of our empirical analysis.

- *Unary (N = 1), UST* A simplified digit-level variant, used as a fundamental model for our theoretical proofs and the "No-Go" theorem.

By establishing these two boundary cases, we can formally demonstrate how gradient stability and token boundaries affect numerical reasoning across disparate scales.

**Triadic Case (**$N = 3$**)** The integer part is split into triads (groups of three digits) from right to left. Each triad receives a suffix that denotes its power of $10^3$: $K$ (thousands, $10^3$), $M$ (millions, $10^6$), $B$ (billions, $10^9$), $T$ (trillions, $10^{12}$), $Q$ (quadrillions, $10^{15}$), and so on. The rightmost triad (units) receives no suffix.

The fractional part is split into triads from left to right. To maintain bijectivity, a *parallel system of replicated markers* is used: the first triad after the decimal point receives the suffix $p$ ($10^{-3}$), the second $pp$ ($10^{-6}$), the third $ppp$ ($10^{-9}$), etc. The decimal point itself is represented by a special token [.].

**Example.** $12\,345\,678.9012$ is tokenized as:

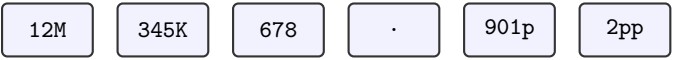

**Unary Case (**$N = 1$**)** In the unary (digit-level) variant each digit becomes a separate token with a positional suffix. For the integer part, digits are processed left to right and receive suffixes $\_ik$ where $k$ decreases from the number of digits down to 1. For the fractional part, suffixes $\_p1, \_p2, \_p3, \ldots$ increase.

**Example.**

$1234.567$ becomes:

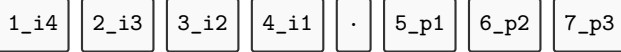

**Vocabulary Size**  We estimate the number of tokens required by both variants for numbers in the range $10^{-15}$ to $10^{18}$. The triadic tokenization in this case gives a total vocabulary of 11000 tokens for TST. The unary tokenization needs 330 tokens.

**Notation and Terminology**

For quick reference, we list the main symbols and abbreviations used throughout the paper.

| Symbol / Term | Meaning |
|---|---|
| NST | N-adic Suffix Tokenization (general family, $N \geq 1$) |
| TST | Triadic Suffix Tokenization ($N = 3$) |
| UST | Unary (digit-level) Suffix Tokenization ($N = 1$), used for theoretical proofs |
| BPE | Byte-Pair Encoding, standard subword tokenization |
| MALL | Magnitude-Aware Log Loss, scalar function $\mathcal{F}_{\text{MALL}}(v, v_{\text{true}})$ |
| $\mathcal{L}_{\text{MALL}}$ | Expected MALL loss over token probabilities (training objective) |
| DNL | Deferred Numerical Loss – MALL applied to hardmax-decoded value (auxiliary) |
| CE | Cross-entropy loss (standard language modelling objective) |
| MSLE | Mean Squared Logarithmic Error |
| MAE / MSE | Mean Absolute / Squared Error |
| L2R | Left-to-right tokenisation order (common in BPE) |
| R2L | Right-to-left tokenisation order |

MALL and $\mathcal{L}_{\text{MALL}}$ are introduced in Section 3, DNL is defined in Section 3.4.

## 3  Magnitude-Aware Log Loss

We define the MALL loss between two numerical values $v, v_{\text{true}} \in \mathbb{R}^+$ as

$$\mathcal{F}_{\text{MALL}}(v, v_{\text{true}}) = \log_{10}\left(1 + \frac{|v - v_{\text{true}}|^{1.2}}{\min(v, v_{\text{true}}) + 0.001 \max(v, v_{\text{true}})}\right) + 0.01 \left|\log_{10}(v) - \log_{10}(v_{\text{true}})\right|^3$$

The specific structure of $\mathcal{F}_{\text{MALL}}$ is governed by several critical design requirements:

1. *Relative Error Sensitivity:* The inclusion of $\min(v, v_{\text{true}})$ in the denominator is essential for creating a "local floor" that forces the model to focus on the relative error within the current scale, ensuring that the mantissa is refined with equal pressure regardless of the order of magnitude.

2. *Zero-Stability Buffer:* The weighted maximum ($0.001 \max$) serves as a stability anchor. It prevents the loss from exploding or behaving erratically near zero. The specific coefficient 0.001 is chosen to balance this stability with the need to concentrate the optimizer's attention on the specific numerical precision (the mantissa or significant digits).

3. *Mantissa Exponent:* The exponent 1.2 provides a "sharper" signal than a linear response ($\alpha = 1$) for small differences, which is necessary for continuity as $v \to 0$. This value also ensures that even at extreme sub-unit scales (e.g., $10^{-6}$), the model still receives a non-negligible gradient for precise mantissa alignment.

4. *Cubic Magnitude Penalty:* The logarithmic term with power $k = 3$ is necessary because the primary mantissa term inherently produces diminishing gradients at large structural distances. Experimentally, the ratio between the 0.01 coefficient and the cubic power ensures that transitions between adjacent orders (e.g., $10^k \to 10^{k+1}$) are smooth and "low-pressure," while simultaneously providing powerful, yet bounded, corrective signals for monumental magnitude errors.

For pairs where $\min(v, v_{\text{true}}) = 0$, the cubic magnitude penalty is omitted. This prevents unbounded growth from $\log_{10}(0)$ and avoids overpenalizing small absolute errors near zero, such as the difference between 1.0000001 and 1.000000 — where the relative error is negligible but the logarithmic penalty would be infinite.

**Choice of Hyperparameters.** The parameters were tested on the number field to provide good gradient value. But along with the formula part, each parameter was chosen so that the corresponding term optimally fulfills its intended role. While these values may benefit from further tuning for specific architectures, the gradient field analysis produced by this function demonstrates they provide a robust and balanced distance metric. The relative weighting between mantissa and magnitude penalty can be further tuned depending on whether the task prioritizes precise mantissa alignment or robust order-of-magnitude estimation. Here we provide a short instruction on tuning the coefficients for some certain needs:

1. *Exponent 1.2*: This is the primary balancing factor between scales. A higher exponent increases sensitivity for macro-scale magnitudes (large numbers) while reducing it for micro-scale magnitudes; it is kept close to 1 to ensure that the precision signal remains actionable across the entire 33-order range and to guarantee continuity at zero distance.

2. *Coefficient 0.001*: Lowering this value widens the "local lens" (allowing the relative error to dominate at smaller scales), while increasing it "blunts" the lens, providing more stability against outliers.

3. *Power 3 and 0.01*: Increasing the power provides a more aggressive "telescopic" correction for order-of-magnitude errors, while the 0.01 coefficient acts as a global gain to prevent this penalty from drowning out the mantissa-focused term.

**Lemma** (Properties of MALL).

The function $\mathcal{F}_{\text{MALL}}$ satisfies:

1. $\lim_{v \to v_{\text{true}}} \mathcal{F}_{\text{MALL}}(v, v_{\text{true}}) = 0$.

2. For all $v, v_{\text{true}} > 0$ on $[10^{-15}, 10^{18}]$: $0 \leq \mathcal{F}_{\text{MALL}} \leq 367$.

*Proof.*

(1) Assume $v_i > v$ without loss of generality. In the first term T1, the denominator is $\min(v_i, v) + 0.001 \max(v, v_i) \geq 0.001 \cdot v_i$. Let the internal ratio of the first term be denoted as the *precision kernel* $\mathcal{K}_1$. Since $v_i \geq |v_i - v|$, we have

$$\mathcal{K}_1 \leq \frac{|v_i - v|^{1.2}}{0.001 \cdot v_i} \leq 1000 \cdot |v_i - v|^{0.2}.$$

As $v_i \to v$, the right-hand side tends to 0, hence the first term $\log_{10}(1 + \mathcal{K}_1) \to 0$. The second term $0.01|\log_{10}(v_i) - \log_{10}(v)|^3 \to 0$ because $|\log_{10}(v_i/v)| \to 0$. Thus $\mathcal{F}_{\text{MALL}}(v_i, v) \to 0$. The case $v > v_i$ is symmetric.

(2) Assume $v_i > v$ without loss of generality. As in (1),

$$\mathcal{K}_1 \leq 1000 \cdot |v_i - v|^{0.2} \leq 1000 \cdot v_i^{0.2}.$$

Since $v_i \leq 10^{18}$, we have $\mathcal{K}_1 \leq 1000 \cdot (10^{18})^{0.2} = 1000 \cdot 10^{3.6} = 10^{6.6}$, hence

$$T1 = \log_{10}(1 + \mathcal{K}_1) \leq \log_{10}(1 + 10^{6.6}) < 7.$$

(3) The maximum logarithmic difference over the range $[10^{-15}, 10^{18}]$ is 33, so the second term satisfies

$$0.01 \cdot |\log_{10}(v_i) - \log_{10}(v)|^3 \leq 0.01 \cdot 33^3 = 359.37.$$

Thus $\mathcal{F}_{\text{MALL}}(v_i, v) < 7 + 359.37 = 366.37$.

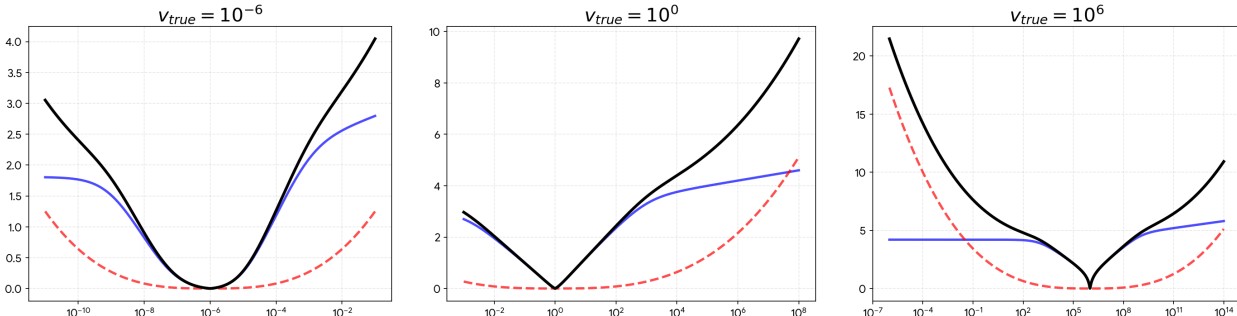

Figure 1: The analytical global landscape of $\mathcal{F}_{\mathrm{MALL}}$ across disparate scales: micro-scale ($v_{\mathrm{true}} = 10^{-6}$), unit scale ($v_{\mathrm{true}} = 10^0$), and macro-scale ($v_{\mathrm{true}} = 10^6$). The blue solid line represents the precision-focused term $T1$, the red dashed line denotes the magnitude penalty $T2$, and the black solid line indicates the total loss value. The invariant sharpness of the local minima and the constant gradient availability across 20 orders of magnitude demonstrate the scale-agnostic stability of the metric.

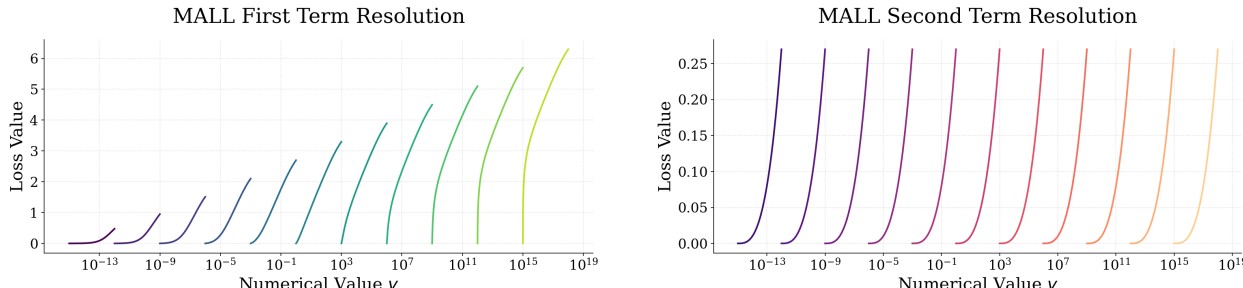

Figure 2: The analytical landscape of $\mathcal{F}_{\mathrm{MALL}}$ first term $T1$ and the second term $T2$ inside the NST triad. In each case we calculate the difference with the smallest token in N-ad.

As illustrated in Figure 1, the MALL loss provides a robust and consistent optimization signal across the $10^{-15}$–$10^{18}$ numerical spectrum. Within the immediate neighborhood of $v_{\mathrm{true}}$, the precision term $T1$ dominates, creating a sharp V-shaped landscape that drives mantissa refinement. As the predicted value deviates beyond a single order of magnitude, $T1$ inevitably enters a logarithmic plateau; however, the cubic magnitude penalty $T2$ simultaneously accelerates, ensuring that the total gradient signal never vanishes. This synergy ensures that the optimizer is guided by high-resolution signals for local errors and by powerful corrective signals for global magnitude misalignments, regardless of the absolute scale of the target value.

Figure 2 illustrates the gradient signals for a pair of values located within a single magnitude triad. It is evident that MALL, with the provided hyperparameters, remains sensitive to both small and large magnitudes while assigning greater weight to larger values.

It is worth noting that, although MALL was originally designed with a focus on a magnitude triad, it demonstrates robust performance globally. When used to generate gradients for training, the metric remains informative and stable regardless of the underlying tokenization scheme. We provide a detailed demonstration of this universal behavior in Section 3.1, where we analyze gradient issues in BPE, and in Section 3.3, through a comparative analysis of NTL (MAE), MALL, and MSLE .

When used for training a language model, the expected loss under the predicted distribution $p$ over token values $v_i$ is

$$\mathcal{L}_{\mathrm{MALL}} = \mathbb{E}_p\big[F_{\mathrm{MALL}}(v, v_{\mathrm{true}})\big] = \sum_i p_i \, F_{\mathrm{MALL}}(v_i, v_{\mathrm{true}}),$$

which is fully differentiable with respect to the logits via the softmax gradient. For tokens representing numbers, it is combined with the standard cross-entropy loss:

$$\mathcal{L}_{\text{total}} = (1 - \alpha) * \mathcal{L}_{\text{CE}} + \alpha \, \lambda \, \mathcal{L}_{\text{MALL}},$$

where $\lambda$ is a balancing coefficient for the $\mathcal{L}_{\text{MALL}}$ itself, and $\alpha$ is a balancing coefficient between categorical and numerical value objectives.

It is important to note that the $\mathcal{L}_{\text{MALL}}$ component is selectively applied only when the target token represents a numerical value. In cases where the target is a standard linguistic token, the model defaults to the standard $\mathcal{L}_{\text{CE}}$. This conditional application ensures that the numerical prior specifically guides the model's behavior in mathematical contexts without interfering with its general language acquisition.

We suggest that the hyperparameter $\alpha$ is beneficial for guiding the neural network to prioritize numerical predictions over non-numerical ones in a numerical context. Without $\alpha$, a "wrong number" prediction would incur a compounded penalty from both $\mathcal{L}_{\text{CE}}$ and $\mathcal{L}_{\text{MALL}}$, potentially producing a larger gradient than a "word instead of a number" prediction. This could inadvertently encourage the model to prefer categorical tokens (words) simply to avoid the additional numerical loss.

However, to simplify hyperparameter tuning, a straightforward formulation using solely the $\lambda$ coefficient may be employed:

$$\mathcal{L}_{\text{total}} = \mathcal{L}_{\text{CE}} + \lambda \mathcal{L}_{\text{MALL}}. \tag{1}$$

As demonstrated in Zausinger et al. (2025), this proves to be effective to combine standard cross-entropy loss with regression loss in practice.

The function $\mathcal{F}_{\text{MALL}}(v, v_{\text{true}})$ defined above serves different purposes depending on how it is used. To avoid confusion, we explicitly distinguish four roles:

Table 1: Four roles of MALL in this paper.

| Role | Definition | Application in paper |
|------|-----------|---------------------|
| MALL metric | $F_{\text{MALL}}(v, v_{\text{true}})$ (scalar, no expectation) | Theoretical analysis, Table 2, loss landscapes (Figs. 1, 2) |
| MALL regression loss (scalar, hard-max) | $F_{\text{MALL}}(v_{\text{pred}}, v_{\text{true}})$ with $v_{\text{pred}}$ decoded from a sequence | Deferred Numerical Loss (DNL) – auxiliary regulariser |
| Gradient over token probabilities | $\mathcal{L}_{\text{MALL}} = \sum_i p_i \, F_{\text{MALL}}(v_i, v_{\text{true}})$ where $p_i$ are softmax outputs | Primary training objective in multi-token experiments (Section 3.3 and Appendix) |
| MALL as auxiliary regulariser | $\mathcal{L}_{\text{total}} = \mathcal{L}_{\text{CE}} + \lambda \mathcal{L}_{\text{MALL}}$ (or with $\alpha$ gating) | Used in Experiment 2; recommended for LLM fine-tuning |

Unless stated otherwise, when we refer to "MALL loss" in the context of training we mean the expected loss $\mathcal{L}_{\text{MALL}}$. The scalar function $\mathcal{F}_{\text{MALL}}$ is called the MALL metric.

## 3.1 The Gradient Problems of BPE Tokenization

To theoretically justify the advantages of *Magnitude-Aware Log Loss* (MALL) combined with the NST or other value bijective tokenization, we analyze the loss landscapes of different approaches. For comparison, we define a baseline metric noted as NTL, which represents the L1-based regression objective used in previous works to optimize BPE-tokenized numbers (Zausinger et al., 2025):

$$\mathcal{L}_{\text{NTL}} = \mathbb{E}_p \big[ |v_i - v_{\text{true}}| \big] = \sum_i p_i |v_i - v_{\text{true}}|.$$

Visualizations (Figures 3, 4, and 5) demonstrate the behavior of the loss functions across nine orders of magnitude (up to $10^9$) without requiring empirical neural network training. For clarity, we visualize the

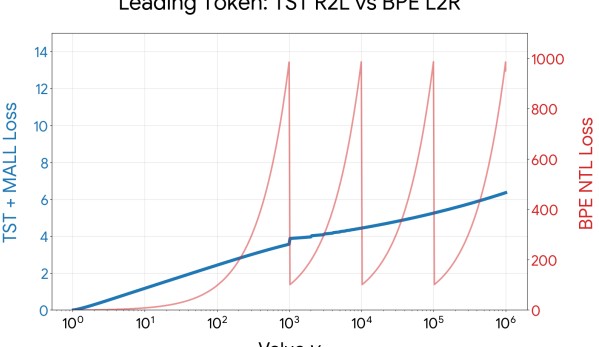

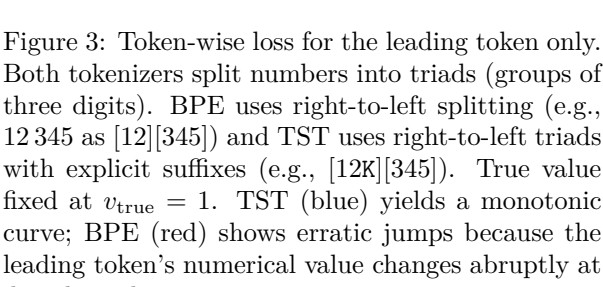

Figure 3: Token-wise loss for the leading token only. Both tokenizers split numbers into triads (groups of three digits). BPE uses right-to-left splitting (e.g., $12\,345$ as [12][345]) and TST uses right-to-left triads with explicit suffixes (e.g., [12K][345]). True value fixed at $v_{\text{true}} = 1$. TST (blue) yields a monotonic curve; BPE (red) shows erratic jumps because the leading token's numerical value changes abruptly at digit boundaries.

Figure 4: Token-wise loss for the leading token: TST (right-to-left triads with suffixes) vs. BPE with left-to-right triads (e.g., $12\,345$ as [123][45]). True value $v_{\text{true}} = 1$. TST remains stable and monotonic. BPE (L2R) causes severe instability because the leading token's value depends on the total number of digits, leading to non-monotonic jumps at boundaries.

functional values (denoted as Loss on the plots), which in our framework directly correspond to the gradient signals acting on the token probabilities. Figures 3 and 4 illustrate how the combination of tokenization and loss behaves specifically for the leading token. Since TST is arranged in triads from right to left (R2L), while BPE may employ various splits (R2L or L2R), we evaluate both variants. For consistency, we assume a triadic split for BPE to illustrate its behavior against the structured TST. Consider the value $12, 345$: a R2L BPE split would be [12][345], while a L2R split would be represented as [123][45]. In TST, the split remains consistently [12$k$][345]. In our analysis, we calculate the gradient relative to the target value 1 and compare only the leading token.

Observations:

- *BPE (L2R)* exhibits an asymmetric and chaotic pattern. In a L2R system, the "head" of the number is inherently unstable; as the number of digits increases, the alignment of subsequent tokens shifts, leading to unpredictable loss spikes.

- *TST (R2L)* naturally aligns with the positional nature of the Arabic numeral system. By anchoring the triadic split to the decimal point, the structural meaning of each token remains constant, resulting in a stable monotonic and mathematically coherent loss curve.

The most critical insight is revealed in Figure 4. In BPE-NTL, as a value transitions across a magnitude boundary (e.g., from $999, 999$ to $1, 000, 000$), the leading token shifts from "999" (thousands) to "1" (millions). Since NTL only considers the mantissa, the loss catastrophically drops from 999 to $\approx 0$. This leads to structural scale blindness: since NTL primarily evaluates the mantissa of the individual token, it fails to distinguish between identical digits at different orders of magnitude. Consequently, the optimizer "perceives" the result as correct, even though the absolute error is monumental. MALL with TST resolves this by incorporating the triadic position into the loss calculation, ensuring that a leading token "1" at the millions scale is penalized far more heavily than a "1" at the units scale.

**Note:** It is important to note that the limitations of the BPE landscape do not preclude the use of magnitude-aware objectives. Specifically, MALL remains fully compatible with BPE-based architectures: by parsing the decoded token sequence into a numerical value, one can leverage MALL as a stable global objective, effectively bypassing the per-token gradient instabilities identified in our analysis.

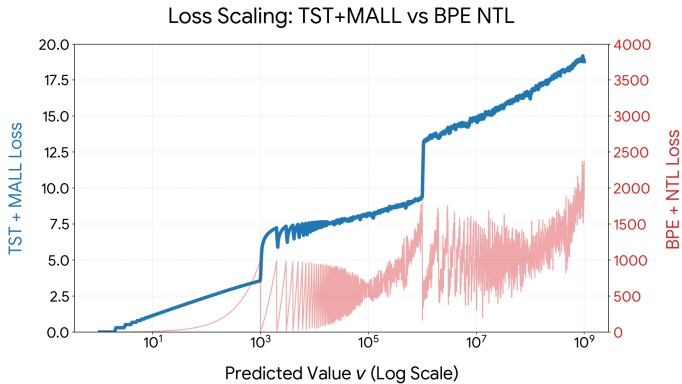

Figure 5: Total aggregated loss across all tokens of a number, with true value fixed at $v_{\text{true}} = 1$ (log scale on x-axis). The loss is the sum of per-token losses over the whole number. A 30-point moving average is applied to suppress high-frequency noise. In both cases the number is split R2L into 3-digit tokens.

While MALL behaves predictably at token boundaries (e.g., ensuring $v_{\text{pred}} = 1000 > v_{\text{true}} = 999$), our analysis reveals that the total loss, when aggregated across all tokens of a number, is not strictly monotonic. Figure 5 illustrates this aggregated loss landscape. To visualize the behavior for both BPE and TST, we compare each numerical value against a fixed target. Note that this visualization employs a 30-point moving average to suppress high-frequency noise and highlight structural variations. While TST+MALL provides a significantly smoother curve than BPE-based tokenization, it still exhibits characteristic "spikes" at major scale transitions.

Our investigation of these spikes however leads to a fundamental observation: the only way to completely eliminate such discontinuities in an additive per-token framework is to employ a strictly linear loss function. In the following Section, we provide a formal proof of this necessity, demonstrating that any continuous additive loss must scale linearly with the magnitude of the number. We then show that this requirement inevitably leads to either gradient explosion or collapse, thereby justifying the rejection of linear additive functions in favor of the MALL framework.

## 3.2 Impossibility of Stable Continuous Additive Per-Token Loss

Next, we consider the following representations of numbers in LLMs, where we have an ordinary decimal notation and the number itself is split into tokens. In the case of NST, the value corresponding to a token is known immediately from the token itself. In the case of BPE, it is assumed that we collect the whole number from the tokens and construct a mapping to a numerical value. We use the decimal system, but this is not a fundamental limitation — the proofs naturally carry over to other bases.

In fact, $F$ computes the distance from 0 to the number and works as a standard loss, i.e., the error is summed as the sum of errors over individual tokens. For example, in the TST case this means that $F(1999) = F([1000][999]) = f(1000, 0) + f(999, 0)$. In BPE notation this could appear as $F(1999) = F([19][99]) = f(1900, 0) + f(99, 0)$. For simplicity, we will write this as $f(k)$ throughout. In the proof we use one-digit UST tokenization. However, since the tokens 0–9 themselves are present in BPE as well, the proof also covers BPE, because we simultaneously establish the same properties for the subset of numbers formed by BPE tokenization.

We prove that any function that decomposes additively across tokens and is continuous in the numerical value necessarily suffers from either gradient explosion or gradient collapse when applied to numerical values spanning $[10^{-15}, 10^{18}]$.

**Setup.** We consider the digit-wise tokenization, where each token represents a single digit 0–9 with an explicit positional suffix. The decimal point is irrelevant for the structure of the argument — it merely separates integer and fractional parts, but the tokenization treats both uniformly.

Let $f : \mathbb{R}^+ \to \mathbb{R}^+$ be a continuous loss function for a single token, with $f(0) = 0$ and $f(1) > 0$. For a number $x$, let $v_1(x), \ldots, v_N(x)$ be the numerical values of its tokens. The total loss is the sum of token-wise losses:

$$F(x) = \sum_{i=1}^{N} f(v_i(x)).$$

We assume $F$ is continuous as a function of $x$.

*Using that $F$ is continuous.* Since the proof looks quite complicated due to induction, we show two steps alone proving that $f(2) = 2f(1)$. This clarifies how the steps in the subsequent inductions are performed.

$$f(1) = F(1) = \lim_{N \to \infty} F\left(\sum_{k=1}^{N} 9 \cdot 10^{-k}\right) = \sum_{k=1}^{\infty} f(9 \cdot 10^{-k})$$

The last step holds because each $9 \cdot 10^{-k}$ is a separate token.

$$f(2) = F(2) = \lim_{N \to \infty} F\left(1 + \sum_{k=1}^{N} 9 \cdot 10^{-k}\right) = f(1) + \sum_{k=1}^{\infty} f(9 \cdot 10^{-k}) = f(1) + f(1) = 2f(1)$$

**Lemma 1** (Transition across an order of magnitude). For any integer $n \geq 0$,

$$f(10^{n+1}) = f(9 \cdot 10^n) + f(10^n).$$

*Proof.* Consider the equality $10^{n+1} = 9.\underbrace{99\ldots9}_{\infty} \times 10^n$. In tokenization, $10^{n+1}$ is a single token [1] at position $10^{n+1}$. The number $9.999\ldots \times 10^n$ consists of:

- one token with value $9 \cdot 10^n$,

- an infinite sequence of tokens with values $9 \cdot 10^{n-1}, 9 \cdot 10^{n-2}, \ldots$

By continuity of $F$ at $10^{n+1}$,

$$f(10^{n+1}) = f(9 \cdot 10^n) + \sum_{k=1}^{\infty} f(9 \cdot 10^{n-k}).$$

The same reasoning applied at scale $10^n$ (to $10^n = 9.999\ldots \times 10^{n-1}$) gives

$$f(10^n) = \sum_{k=1}^{\infty} f(9 \cdot 10^{n-k}).$$

Substituting, we obtain

$$f(10^{n+1}) = f(9 \cdot 10^n) + f(10^n).$$

**Lemma 2** (Linearity on the same order). For any digit $d \in \{0, \ldots, 9\}$ and any scale factor $\delta = 10^n$,

$$f(d \cdot \delta) = d \cdot f(\delta).$$

*Proof by induction on $d$.* Base $d = 0$: $f(0) = 0$. Assume $f((d-1)\delta) = (d-1)f(\delta)$. Then consider the transition $(d-1).999\ldots \times \delta \to d\delta$:

$$f(d\delta) = f((d-1)\delta) + \sum_{k=1}^{\infty} f(9 \cdot 10^{-k}\delta).$$

By Lemma 1 at scale $\delta$, the sum equals $f(\delta)$. Hence $f(d\delta) = (d-1)f(\delta) + f(\delta) = df(\delta)$.

**Lemma 3** (Propagation across orders). For any integer $n \geq 0$, $f(10^n) = 10^n f(1)$.

*Proof by induction on n.* Base $n = 0$: $f(1) = 1 \cdot f(1)$. Assume $f(10^n) = 10^n f(1)$. Then by Lemma 1 and Lemma 2,

$$f(10^{n+1}) = f(9 \cdot 10^n) + f(10^n) = 9f(10^n) + f(10^n) = 10 \cdot f(10^n) = 10^{n+1} f(1).$$

**Note** (Generalization to arbitrary tokenizations). The induction proceeds on two independent levels.

- *Between tokens (order transition).* Lemma 1 only requires the ability to approach a power of ten from below by an infinite sequence of tokens with digit 9. This holds for any base-10 tokenization regardless of how digits are grouped.

- *Inside a token (linearity on the same order).* Lemma 2 uses the fact that adding one unit at a given digit can be expressed by pushing a sequence of 9's to the next token (or to the next digit inside the same token). Hence the induction applies irrespective of the token boundary positions.

After inference, a BPE tokenization is decoded to the final numerical value by concatenating the digit sequences of the generated tokens. At this stage the number is a concrete real value, and the loss is computed on that final number. Consequently, the induction argument applies directly to the decoded number, and the No-Go Theorem remains valid for any base-10 tokenization, including BPE, as long as the loss is evaluated on the fully reconstructed number.

**Theorem** For any additive continuous per-token loss $F$ on the range $[10^{-15}, 10^{18}]$, either gradient explosion or gradient collapse occurs.

*Proof.* From Lemma 3, $f(10^{18}) = 10^{18} f(1)$. If $f(1) \geq 1$, then $f(10^{18}) \geq 10^{18}$ — explosion. If $f(1) < 1$, to keep $f(10^{18}) \leq 1000$ we need $f(1) \leq 10^{-15}$. Then $f(10^{-15}) = 10^{-15} f(1) \leq 10^{-30}$ — collapse below machine precision.

The following corollary is intuitively clear from the theorem: a continuous additive function must be linear; on a 33-order range, linearity leads to explosion or collapse. Thus, to remain bounded, the function must have discontinuities. If all jumps were positive, the function would grow even faster, so a downward jump is unavoidable. Below we give a rigorous proof that this holds for any monotone token-additive function.

**Corollary**(Paradox of Monotonicity Counter-Intuition) To satisfy the boundedness and precision constraints established in the No-Go Theorem on $[10^{-15}, 10^{18}]$, any alternative non-continuous additive per-token loss function $F_{mon}(x)$ cannot be globally monotonically increasing. To prevent gradient explosion without suffering from collapse, the function is mathematically forced to exhibit downward discontinuous jumps across magnitude transitions.

*Proof.* Assume for contradiction that there exists a strictly monotonically increasing, non-continuous additive function $F_{mon}(x)$ that remains bounded. Since $F_{mon}$ is monotonic, any discontinuity at a point $x_0$ must be a jump of positive magnitude: $\Delta(x_0) = \lim_{x \to x_0^+} F_{mon}(x) - \lim_{x \to x_0^-} F_{mon}(x) > 0$. We construct a corrected function $F(x)$ by systematically subtracting the cumulative sum of these positive jumps at each discontinuity at the level of the most-significant token, preserving additivity, effectively shifting the graph downward to eliminate the gaps. The correction is performed from right to left: starting with the smallest order $(10^{-15})$ and moving toward larger ones $(10^{-14}, 10^{-13}, \ldots, 10^{18})$. This order ensures that modifying a more significant token (higher order) does not reintroduce discontinuities into already smoothed tails of lower orders. Outside this interval, on $[0, 10^{-15})$, we define $F$ as $F(x) = \dfrac{F_{\text{mon}}(10^{-15})}{10^{-15}} \cdot x$ so that it is continuous at the boundary $10^{-15}$ and per-token additive. This extension does not affect any argument on the main interval. The resulting function $F(x)$ is continuous and satisfies $F_{mon}(x) \geq F(x)$ for all $x > 10^{-15}$. Moreover,

by construction, for any $x < y$ we have $|F(y) - F(x)| \leq |F_{mon}(y) - F_{mon}(x)|$. Since $F(x)$ is continuous and additive, by Lemma 3 it must scale linearly as $f(10^n) = 10^n f(1)$, leading to exponential gradient explosion at $10^{18}$. Because $F_{mon}(x) \geq F(x)$ and for any $x < y$ we have $|F(y) - F(x)| \leq |F_{mon}(y) - F_{mon}(x)|$ the original monotonic function must explode as well, yielding a contradiction. Thus, $F_{mon}$ must exhibit downward jumps, violating global monotonicity.

**Remark** Specifically, to prevent exponential gradient explosion across scaling ranges, the loss function is mathematically forced to exhibit downward discontinuous jumps. While this does not imply that a downward jump must occur at every single boundary point (such as exactly at $1999 \rightarrow 2000$), such downward discontinuities must systematically exist across major magnitude transitions for the function to remain bounded, forcing the architecture to topologically treat larger values as states of lower penalty ($F(x_{high}) < F(x_{low})$).

Consequently, we distinguish between two types of the "sawtooth" effect:

- *BPE-sawtooth*: Results from a complete lack of order awareness within the token vocabulary, leading to stochastic loss jumps.

- *Loss-sawtooth*: Arises because any function capable of seamlessly "stitching" token boundaries is inherently unsuitable for gradient-based optimization due to its super-linear growth.

*Note on MALL, Spikes, and Softmax:*

This theorem formally proves that any loss function capable of providing a stable, non-vanishing gradient signal across multiple scales must inherently exhibit "spikes" or discontinuities at token boundaries.

However, as demonstrated in Section 3.4, such spikes should not be viewed as a deficiency of the metric. Instead, they represent an essential regularization feature that addresses the inherent limitations of the Softmax function in discrete numerical spaces. From the perspective of numerical integrity, providing a small error at boundary transitions would be a fundamental mistake, as it would fail to penalize the emergence of numerically distant "phantom" sequences.

## 3.3 Gradient Field Analysis: MALL vs. MSLE and MAE

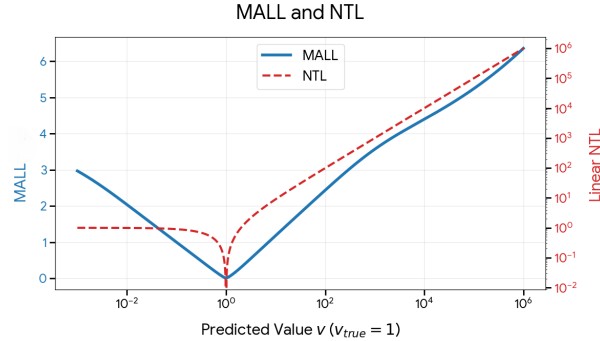
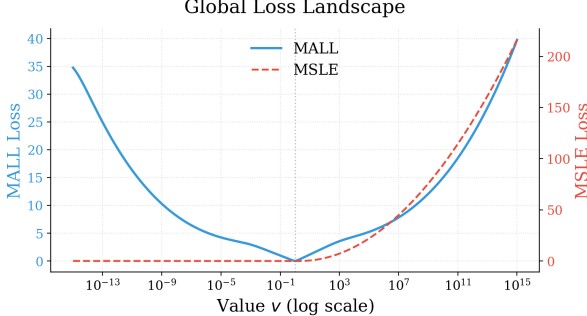

Figure 6: MALL and NTL (MAE) Global

Figure 7: MALL and MSLE

Standard MSE exhibits even more aggressive scaling than MAE (L1,NTL), further exacerbating the vanishing gradient problem for small-scale values while simultaneously triggering immediate gradient explosion at larger magnitudes. Consequently, we exclude MSE that is used for xVal from the detailed graphical analysis to maintain visual clarity and focus on informative gradients. However, for a complete comparison of scale instability, we include its values in the general analysis (Table 2).

Instead we compare our approach with Mean Squared Logarithmic Error (MSLE), a popular scale-robust metric implemented in frameworks like Scikit-Learn Pedregosa et al. (2011) and Keras Chollet et al. (2015).

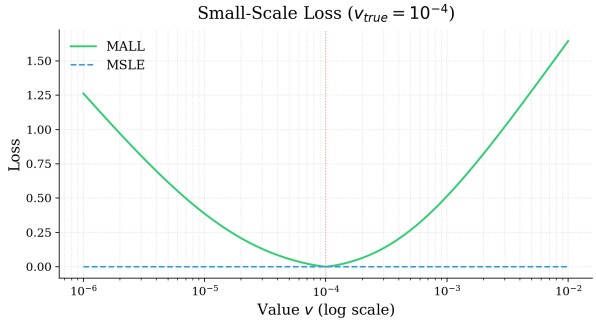
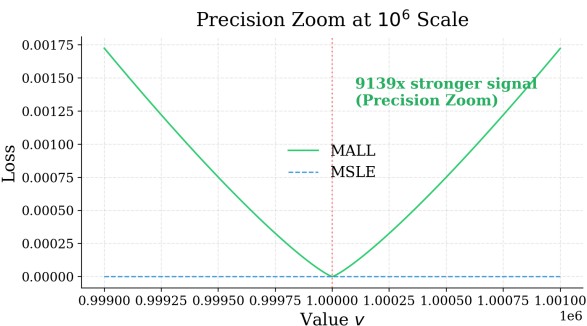

Figure 8: MALL and MSLE on micro numbers      Figure 9: MALL and MSLE on big numbers

The MSLE metric for two numbers is defined as:

$$\mathcal{F}_{\mathrm{MSLE}}(v, v_{\mathrm{true}}) = (\log(1 + |v|) - \log(1 + |v_{\mathrm{true}}|))^2.$$

This quadratic loss on log-transformed values with a $+1$ offset ensures stability near zero but introduces the limitations discussed below: sub-unit blindness ($\log(1 + \epsilon) \approx \epsilon$) and mantissa erosion at large scales.

As illustrated in Figure 6 and Figure 7, the contrast in gradient stability is most evident at scale extremes, where MALL's dynamic normalization prevents the vanishing and exploding gradient problems inherent in fixed-shift metrics. The main problems of baseline metrics:

1. *The Scale Dominance Problem (L1 NTL)*: Standard Linear NTL is intrinsically ill-conditioned for wide-range regression. For instance, a 20x relative error between $10^{-6}$ and $2 \times 10^{-5}$ produces a negligible absolute loss ($1.9 \times 10^{-5}$), effectively starving the model of any meaningful gradient signal. Conversely, similar relative errors at large scales (e.g., $10^8$ vs $2 \times 10^9$) generate gradient magnitudes of order $O(10^9)$, leading to immediate numerical instability and catastrophic weight divergence Figure 6.

2. *Bi-directional Insensitivity and Precision Collapse of MSLE*: While MSLE employs a logarithmic transform, its reliance on a fixed $+1$ offset and a shallow quadratic derivative leads to a critical loss of resolution at both extremes of the numerical spectrum:

   - *Sub-unit Blindness*: For $v \ll 1$, MSLE creates a near-zero loss plateau where it becomes insensitive to order-of-magnitude shifts, as $\log(1 + \epsilon) \approx \epsilon$ (see Fig. 8 ).
   - *Mantissa Erosion*: At large scales, MSLE fails to maintain relative precision. Analytical probing (see Fig. 9) reveals that MSLE effectively fails to distinguish between values like $1,001,000$ and $1,000,000$, treating a thousand-unit discrepancy as numerical noise. This prevents the model from accurately anchoring mantissa precision within high-magnitude ranges.

MALL transcends these limitations through three distinct mechanisms that transform numerical optimization into a structured coordinate descent across the embedding manifold:

1. *Landscape Resolution via Dynamic Denominator*: Instead of a fixed offset, MALL utilizes a dynamic denominator: $\min(v, v_{\mathrm{true}}) + 0.001 \max(v, v_{\mathrm{true}})$. This preserves a sharp, V-shaped loss landscape across the entire 33-order spectrum. By maintaining scale-invariant sensitivity, MALL ensures that a 20x error produces a consistent, actionable signal regardless of whether the target is $10^{-12}$ or $10^{12}$.

2. *High-Precision Mantissa Refinement*: Unlike MSLE, which treats mantissa errors as negligible residuals for large numbers, MALL's $|v - v_{\mathrm{true}}|^{1.2}$ term ensures the landscape remains strictly "pointed" at the ground truth. This provides the necessary gradient tension for the model to refine digit-level precision even at astronomical scales, preventing the precision collapse seen in fixed-log metrics.

3. *Cubic Order-of-Magnitude Penalty*: To prevent "magnitude hallucinations," MALL introduces a cubic penalty term: $0.01 \cdot |\log_{10}(v) - \log_{10}(v_{\text{true}})|^3$. Unlike the shallow parabola of MSLE, the cubic term grows aggressively as the predicted exponent deviates. On one hand, this forces architectural updates to prioritize correct order-of-magnitude alignment before fine-tuning the mantissa, mirroring a human-like approach to numerical estimation; on the other hand, the regulation coefficient of 0.01 allows the first term to take the main part when magnitudes are close.

Through these mechanisms, MALL effectively transforms the optimization task from a simple value regression into a robust search for both magnitude and precision. Classical robust losses, such as the Huber loss (Huber, 1964) or its differentiable analogue log-cosh (Chen et al., 2019), combine MAE and MSE behavior via a threshold $\delta$ and are designed to reduce the influence of outliers. However, they operate within a fixed error scale and do not address the explosion or collapse of gradients across 30 orders of magnitude. One could reinterpret MALL as an adaptive robust loss: the first term $T1$ handles local precision (similar to a quadratic/linear switch but scaled by the current magnitude), while the cubic term $T2$ acts as a penalty on gross magnitude errors. Nevertheless, MALL does not rely on an explicit threshold and remains stable without additional tuning.

Table 2: Comparative Analysis of Loss Metrics.

| $v_{\text{true}}$ | $v_{\text{pred}}$ | MSE | MAE (NST) | MAE (BPE) | MSLE | MALL T1 | MALL T2 | Full MALL |
|---|---|---|---|---|---|---|---|---|
| $10^{-12}$ | 999 | $1.0 \times 10^6$ | $1.0 \times 10^3$ | 998 | 9.000 | 3.600 | 33.747 | 37.347 |
| $10^{-6}$ | $2 \times 10^{-5}$ | $3.6 \times 10^{-10}$ | $1.9 \times 10^{-5}$ | 19 | 0.000 | 0.494 | 0.022 | 0.516 |
| 0.01 | 0.05 | $1.6 \times 10^{-3}$ | 0.04 | 40 | 0.001 | 0.490 | 0.003 | 0.493 |
| 0.01 | $10^8$ | $1.0 \times 10^{16}$ | $1.0 \times 10^8$ | 90 | 63.931 | 4.600 | 10.000 | 14.600 |
| 200 | 500 | $9.0 \times 10^4$ | $3.0 \times 10^2$ | 300 | 0.157 | 0.755 | 0.001 | 0.756 |
| $10^{-12}$ | 200 | $4.0 \times 10^4$ | $2.0 \times 10^2$ | 199 | 5.305 | 3.460 | 29.248 | 32.708 |
| $10^8$ | $2 \times 10^9$ | $3.6 \times 10^{18}$ | $1.9 \times 10^9$ | 98 | 1.693 | 3.126 | 0.022 | 3.148 |
| $10^{11}$ | $1.01 \times 10^{11}$ | $1.0 \times 10^{18}$ | $1.0 \times 10^9$ | 1 | 0.000 | 0.211 | 0.000 | 0.211 |
| 200 | 199 | 1.0 | 1.0 | 1 | 0.000 | 0.002 | 0.000 | 0.002 |

To further quantify these observations, we present a comparative analysis of gradient values across diverse numerical regimes in Table 2. These examples numerically demonstrate the failure modes of standard metrics while highlighting the balanced behavior of MALL. Specifically, the table illustrates how the two components of MALL interact: the term T1 effectively captures mantissa discrepancies, maintaining sensitivity at both micro and macro scales, while the term T2 dominates when order-of-magnitude errors occur. This dual-mechanism ensures that the model receives a consistent gradient signal for both precision refinement and magnitude alignment, a property clearly absent in MSLE, MSE and MAE baselines. For instance, at a macro scale ($10^{11}$ vs. $1.01 \times 10^{11}$), despite an absolute error of $10^9$, both MSLE and the pure log-loss component T2 register exactly 0.000, going completely blind to the mantissa discrepancy. However, T1 rescues the optimization by providing a robust signal of 0.211. Similarly, at micro scales ($10^{-6}$ vs. $2 \times 10^{-5}$), MSLE collapses to 0.000 due to its +1 shift, and T2 yields an insufficient 0.022, while T1 generates a strong value of 0.494. These specific cases prove that a pure log-loss T2 or shifted metrics are inadequate for fine-grained precision, making the local component T1 strictly essential.

### 3.4 Softmax Limitations and Deferred Numerical Loss

Our per-token numerical loss works directly with the probability distributions and provides a sensible gradient signal in most cases. Consider a situation where the model assigns independent probabilities to tokens at different positions. For example, suppose the true number is 2000 (split into tokens [2K] or [2] and [000] ) and the model recognizes that the numerical values 1999 and 2000 are very close. As a result, the model produces the following distributions:

$$p_1(1) = 0.5, \ p_1(2) = 0.5, \qquad p_2(000) = 0.5, \ p_2(999) = 0.5.$$

Because the positions are treated independently, this joint distribution implies that the model considers four sequences with equal probability 0.25: 1999, 2999, 1000, and 2000. Among these, the true number 2000

has probability 0.25, the relatively close number 1999 also has probability 0.25, while the remaining two sequences (2999 and 1000) have a cumulative probability of 0.5 and are numerically far from the true value Figure 10. Consequently, the expected numerical error over the joint distribution is large. Our per-token loss correctly detects this uncertainty and produces a large loss (approximately 3.8), signalling the model that it must resolve the ambiguity. We name this a Softmax boundary problem since the loss in this case is not the exact metric or an explicit tokenization mistake. Such a discontinuity (the Softmax boundary problem) occurs at all transitions where a lower token accumulates into a higher one — that is, when the order of magnitude changes or when the value in a higher token "flips" (e.g., from 1 to 2 after 999). In the proof of the No-Go theorem, every step uses exactly such discontinuity points.

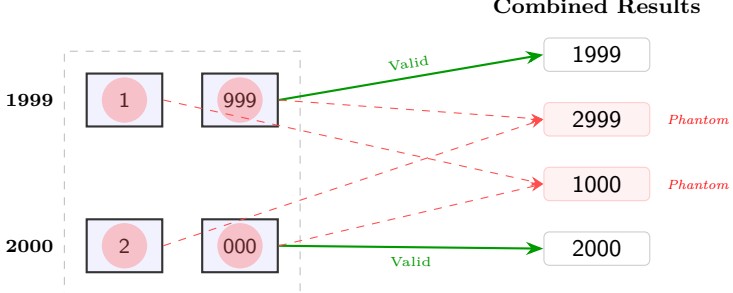

Figure 10: Softmax Boundary Problem: When the model is uncertain between two valid numbers (e.g., 1999 and 2000), independent sampling allows the head of one to mix with the tail of the other, resulting in catastrophic errors.

*This example highlights two points:*

1. *Consistency Constraint:* If the target is 2000, the model cannot be allowed to assign high probabilities to both "1" (at the thousands position) and "999" simultaneously. Doing so creates a massive numerical error, even if each token seems "reasonable" in isolation.

2. *Gradient as a Catalyst:* Consequently, the large gradient produced by the MALL or other metric at these boundaries is not a flaw. On the contrary, it serves as a necessary signal that forces the Softmax to make a decisive, unambiguous choice. This prevents the emergence of "phantom" results that are numerically distant from the true value.

The only way to reduce this loss is for the model to assign high probability to a single consistent sequence. This effectively means making a hard decision (argmax) at each position. However, once hard decisions are made, the natural loss is the deferred numerical loss (DNL) for the entire list of tokens for one number. The idea is to decode the entire predicted sequence into a scalar number and then apply a regression loss. Let the model generate a sequence of tokens $t_1, t_2, \ldots, t_k$ (most significant first). Using the bijective TST mapping, we compute the numerical value of the sequence:

$$V_{\text{pred}} = \sum_{i=1}^{k} v(t_i) \cdot w_i^{(\text{dec})},$$

Where $w_i^{(\text{dec})}$ are the positional weights for decoding (e.g., $w_1^{(\text{dec})} = 1000^{k-1}, \ldots, w_k^{(\text{dec})} = 1$ for integer numbers). In the case of a fixed, fully generated sequence (*hardmax*), the total loss is determined by the MALL function:

$$\mathcal{L}_{\text{MALL}} = \mathcal{F}_{\text{MALL}}(V_{\text{pred}}, V_{\text{true}}).$$

This loss is exact but is defined only after the full sequence has been generated. A key challenge in back-propagating this error is the ambiguity of the digit representation: for numbers near power-of-ten boundaries

(e.g., 999 vs. 1000), it is mathematically non-trivial to determine which specific token caused the magnitude shift. Due to this structural ambiguity, we cannot uniquely attribute the error to a single position.

Instead, we distribute the global MALL signal uniformly across all $N$ tokens that compose the number. The gradient for each token $t_i$ is thus defined as:

$$\frac{\partial \mathcal{L}_{\text{MALL}}}{\partial p_i} = \text{sign}(V_{\text{pred}} - V_{\text{true}}) \cdot \frac{1}{N} \mathcal{F}_{\text{MALL}}(V_{\text{pred}}, V_{\text{true}}).$$

In this formulation, the MALL value acts as a global magnitude for the update, enabled for end-to-end training via a straight-through estimator. While the MALL metric is inherently sensitive to minute differences (e.g., 0.0001 vs 0.00001), any scalar regression approach faces a structural limitation: when a number is collapsed into a single scalar value, errors in the least significant digits inevitably become numerically "invisible" against a larger total, such as 1000.0001 vs 1000.00001. This is not a deficiency of MALL itself, but a consequence of scalar aggregation, which drowns out precision at large magnitudes.

Tokenization addresses this by preserving independent gradient pathways for each positional component. To bridge the gap between these independent tokens and numerical continuity, we introduce the Deferred Numerical Loss (DNL) as a structural safeguard. To balance global consistency and per-token precision, DNL is implemented as an auxiliary loss with a small weight $\lambda \ll 1$. In this setup, DNL acts as a global anchor, providing a coarse correction signal to resolve major "phantom" errors and preventing the model from drifting into numerically distant hallucinations during periods of high softmax uncertainty.

Crucially, DNL must complement rather than replace per-token learning. At major scale transitions (e.g., $1.99\ldots9$ vs $2.00\ldots0$), even a robust global signal can be insufficient to trigger the synchronized symbolic "flip" of all tokens. Thus, the per-token MALL remains the primary driver for discrete, fine-grained convergence, while DNL ensures the model's global numerical intuition remains stable.

## 4 Geometric Order Constraints and Triangle Loss

We propose a Triangle Loss that encourages each embedding $E(T_n)$ to align with the segment connecting its neighbors $E(T_{n-1})$ and $E(T_{n+1})$. Specifically, we penalize the squared distance from $E(T_n)$ to that segment. Unlike using MALL at the end point of the LLM inference, which is possibly to apply for the BPE tokenization, this attempt is possible only in the case if there is a bijection between the token and the token value as is in NST tokenization. In the BPE case the token [111] could represent 111 or 111000 so it's impossible to say how it should be located comparing to tokens [110] and [112] in the embedding space.

$$d(E(T_n), E(T_{n-1})) < d(E(T_{n-1}), E(T_{n+1})), \quad d(E(T_n), E(T_{n+1})) < d(E(T_{n-1}), E(T_{n+1})),$$

where $d(\cdot, \cdot)$ is the Euclidean distance.

For each triple of consecutive tokens $(T_{n-1}, T_n, T_{n+1})$, let $A = E(T_{n-1})$, $B = E(T_{n+1})$, $X = E(T_n)$. We measure how much $X$ deviates from the line segment between $A$ and $B$ using the squared distance to the segment. Define the projection of $X$ onto the line $AB$:

$$t = \frac{(X - A) \cdot (B - A)}{\|B - A\|^2}, \qquad P = A + \max\big(0, \min(1, t)\big)(B - A).$$

If $\|B - A\| = 0$, set $P = A$. The violation for this triple is $d_n = \|X - P\|^2$.

As illustrated in the Figure 11, Triangle Loss acts as a low-pass filter on the numerical manifold. It does not collapse embeddings into a straight line, but rather penalizes local 'shattering' of the embedding space, ensuring that numerical progression remains topologically coherent.

*Addressing Data Sparsity* (Rare Tokens). The primary motivation for Triangle Loss is to provide a structural prior for rare numerical tokens. In large-scale training, tokens representing specific high-magnitude triads (the "long tail" of the distribution) may receive insufficient gradient updates to form a coherent placement

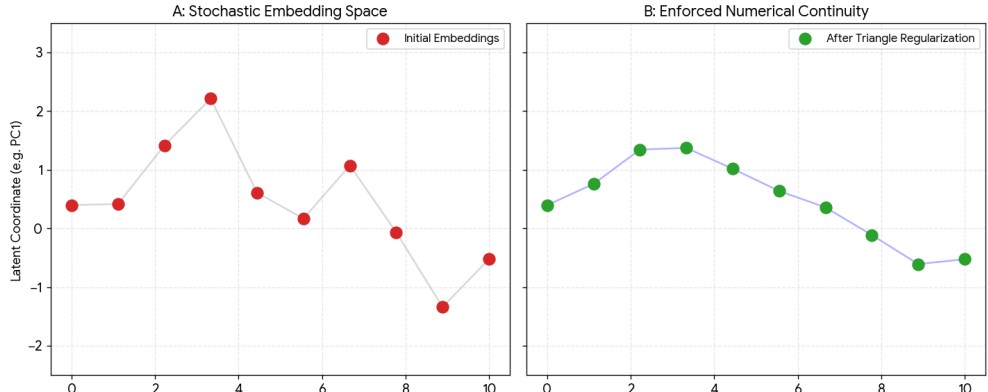

Figure 11: Triangle Loss Regularization

in the embedding manifold. This ensures that a rare token's embedding does not drift too far from its numerical neighbors in the manifold. In this sense, Triangle Loss acts similarly to weight regularization (such as $L_2$) in neural networks: just as weight decay prevents weights from exploding without forcing them to zero, Triangle Loss prevents embeddings from "wandering" into isolated regions of the space. It provides a stabilizing constraint that maintains a baseline level of ordinal coherence, ensuring the token remains part of the global numerical progression while still preserving the unique high-dimensional features learned from its specific contexts.

*Post-processing variant (default).* After training, we can apply a small correction to each violating token without further optimisation. For each $T_n$, we move its embedding slightly towards its projection $P$:

$$E_{\text{new}}(T_n) = (1 - \alpha)E(T_n) + \alpha P,$$

with a small step size (e.g., $\alpha = 0.1$). This operation is repeated a few times. It does not require any gradients and can be used as a final "repair" step to improve the ordinal quality of the embedding space. For the TST vocabulary (11 000 tokens) this yields approximately 11 000 triples; for the unary case (330 tokens) there are about 328 triples, which is negligible.

*Regularisation variant (optional).* The same violation can be added as a differentiable loss during training:

$$\mathcal{L}_{\text{geo}} = \sum_n d_n,$$

where the sum runs over all interior tokens (about 11 000 for TST, 328 for unary). This loss is then combined with the main language modelling objective:

$$\mathcal{L}_{\text{total}} = \mathcal{L}_{\text{LM}} + \lambda \mathcal{L}_{\text{geo}}.$$

This variant requires no additional post-processing but may interfere with non-numerical tokens; we therefore recommend the post-processing method as the default.

## 5    Conclusion

In this paper, we have analytically demonstrated that the failure of modern language models in numerical reasoning is not merely a capacity issue, but a fundamental flaw of the underlying optimization landscape. By examining the gradient fields of existing metrics, we identified two critical bottlenecks: the 'sawtooth' instability of BPE tokenization and the non-invariant precision of current regression losses, which leads to

a critical loss of resolution for sub-unit magnitudes and a failure to distinguish proximal values in high-magnitude ranges.

Our theoretical framework provides a three-fold solution to these challenges. First, MALL (Magnitude-Aware Log Loss) establishes a stable, scale-invariant objective that preserves both global magnitude alignment and high local mantissa precision across at least 33 orders of magnitude on $\mathbb{R}^+$. Importantly, MALL is representation-agnostic: it works with any positive numerical representation that allows a bijective mapping between tokens and numerical values, whether through N-adic Suffix Tokenization, decoded to number BPE sequence, scalar regression, or as a standalone baseline for numerical comparison across the $\mathbb{R}^+$ domain. Second, we establish a No-Go Theorem proving that any continuous additive per-token loss is mathematically incompatible with numerical stability over large ranges. This leads to our analysis of the Softmax boundary problem—a structural failure where independent positional distributions produce numerically distant "phantom" results. We demonstrate that the structured discontinuities in the MALL gradient field act as a necessary catalyst, forcing global consistency across positional tokens where continuous objectives would otherwise collapse. To stabilize this behavior, we propose a deferred global loss with hardmax as a regularization strategy. Furthermore, we introduce Triangle Loss, a geometric regularizer that enforces numerical continuity within the embedding manifold and alleviates the rare-token problem at extreme scales.

Together, these components transform numerical learning from a chaotic search in an ill-conditioned space into a structured coordinate descent. While this work remains primarily theoretical, the analytical evidence suggests that adopting magnitude-aware objectives and structurally sound tokenization is a mandatory prerequisite for achieving robust and coherent numerical intelligence in neural architectures.

## Broader Impact Statement

This work focuses on a technical improvement for numerical optimization in language models. While not directly deployed as a standalone system, the proposed MALL loss and related components could be integrated into larger models used in scientific, financial, or engineering applications. As with any machine learning model, improved numerical fluency does not guarantee correctness or calibration. Users should apply appropriate validation, uncertainty estimation, and domain-specific checks before relying on model outputs in high-stakes decisions. The authors do not foresee specific negative societal impacts beyond the general risks of over-trust in automated systems.

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

# A   Appendix

## A.1   Experiments

**Experiment 1: Regression on classes - predicting only the order of magnitude (33 classes).** Classes represent pure powers of ten $10^e$ with $e \in [-15, 18]$ (33 classes). Target: $10^{-6}$. A small MLP outputs a softmax distribution over the 33 classes. The loss is $\mathbb{E}[\text{distance}] = \sum_i p_i \cdot d_i$ where $d_i$ is a pre-computed distance between the predicted exponent $e_i$ and the true exponent $-6$. Four distance (gradient) functions are compared: MALL, MSE, MAE, MSLE. In all cases the model was trained for 500 epochs with the learning rate of 0.01. The experiment tests how each distance shapes the final probability distribution over orders of magnitude. MSE and MAE are dominated by huge numbers (e.g., $10^{18}$), producing flat or skewed distributions with non-negligible mass on far-away exponents. MSLE compresses the scale but still allows noticeable probability at extremes because its penalty grows too slowly for large logarithmic deviations. Only MALL yields a sharply peaked distribution concentrated around the true order $10^{-6}$ (and its immediate neighbours $10^{-5}$, $10^{-7}$), with virtually zero probability for distant orders like $10^{15}$ or $10^{-15}$. Thus, the experiment isolates the order-of-magnitude problem and demonstrates that a properly structured distance is essential for numerical tasks.

**Experiment 2: multi-token sum prediction (4 tokens, 310 classes).** Each token is a class with $m \in \{0, \dots, 9\}$, $e \in [-15, 15]$ (310 classes). Target 0.001 is represented as the sum of four distinct tokens: $(1, -3)$, $(0, 0)$, $(0, 1)$, $(0, 2)$. A shared backbone with four independent heads (non-autoregressive, like a multi-label classifier) predicts all four tokens in parallel. The loss is the sum over tokens of the expected loss using the same three per-token distances (CE, CE+MALL, CE+MAE$_{\text{mantissa}}$). Metrics: average probability of the correct token class, average argmax accuracy, and the regression error $|\mathbb{E}[\text{sum}] - 0.001|$ where $\mathbb{E}[\text{sum}] = \sum_{\text{tokens}} \mathbb{E}[v_{\text{token}}]$. The test calculates the average results of 10 runs. CE again spreads minuscule probability over astronomically large values (up to $9 \times 10^{15}$), causing $\mathbb{E}[\text{sum}]$ to stay around 10 instead of 0.001 after 20 epochs. CE+MALL concentrates the residual probability near the target order of magnitude ($10^{-4}$–$10^{-2}$), yielding $\mathbb{E}[\text{sum}]$ extremely close to 0.001. CE+MAE$_{\text{mantissa}}$ penalises only wrong mantissa - similar to NTL.

As shown in Figure 13 with $\lambda = 0.02$ and Figure 14 with $\lambda = 0.002$, additional regression loss makes it slightly longer to converge to the exact class but the predicted values are reached considerably faster. This happens because CE itself doesn't see the difference between values and even relatively small probabilities of wrong numbers lead to big value errors. We illustrate it in the Table 3, which shows the top-10 most

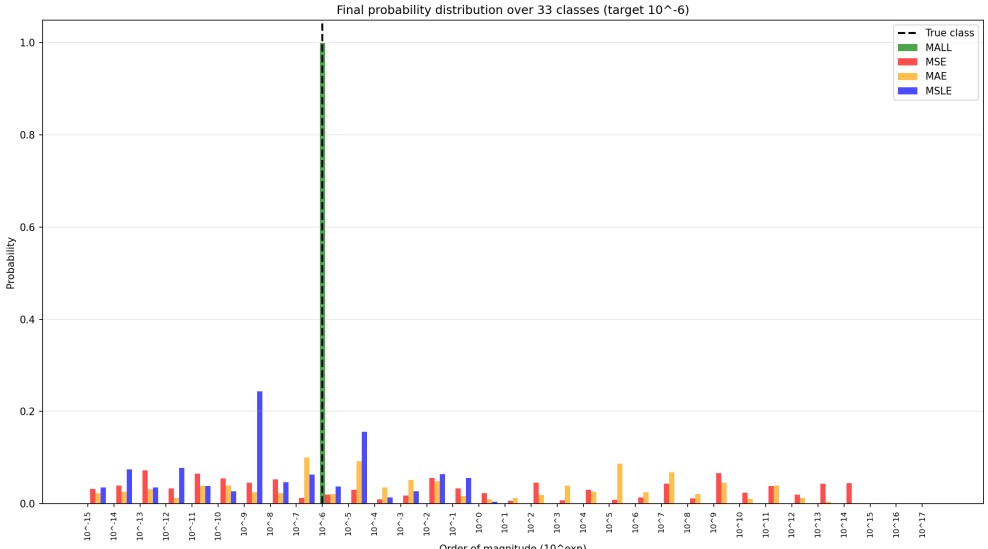

Figure 12: MALL, MSE, MAE, MSLE as a regression on classes.

possible classes for CE and CE with MALL. Considering the second possible class it means that pure CE says: "Your interest rate is 0.1%, if not, then maybe $2.000000 \times 10^{16}\%$". With the use of MALL it changes to: "Your interest rate is 0.1%, if not, then maybe 0.2%".

Table 3: Top 10 most probable classes (CE vs MALL+CE). Target class: (1, -3) with value $10^{-3} = 0.001$.

| Rank | CE | | | MALL+CE | | |
|---|---|---|---|---|---|---|
| | Probability | Value | (m, e) | Probability | Value | (m, e) |
| 1 | 1.000000 | $1.000000 \times 10^{-3}$ | (1,-3) | 1.000000 | $1.000000 \times 10^{-3}$ | (1,-3) |
| 2 | $2.428101 \times 10^{-19}$ | $2.000000 \times 10^{14}$ | (2,14) | $2.069120 \times 10^{-15}$ | $2.000000 \times 10^{-3}$ | (2,-3) |
| 3 | $2.095215 \times 10^{-19}$ | $6.000000 \times 10^{-4}$ | (6,-4) | $1.563666 \times 10^{-15}$ | $6.000000 \times 10^{-4}$ | (6,-4) |
| 4 | $1.825125 \times 10^{-19}$ | $7.000000 \times 10^{-2}$ | (7,-2) | $2.119633 \times 10^{-16}$ | $4.000000 \times 10^{-4}$ | (4,-4) |
| 5 | $4.452810 \times 10^{-20}$ | $3.000000 \times 10^{-13}$ | (3,-13) | $1.675847 \times 10^{-16}$ | $2.000000 \times 10^{-5}$ | (2,-5) |
| 6 | $3.567330 \times 10^{-20}$ | $9.000000 \times 10^{11}$ | (9,11) | $1.545477 \times 10^{-16}$ | $3.000000 \times 10^{-4}$ | (3,-4) |
| 7 | $2.948600 \times 10^{-20}$ | $6.000000 \times 10^{6}$ | (6,6) | $1.507030 \times 10^{-16}$ | $7.000000 \times 10^{-4}$ | (7,-4) |
| 8 | $2.624331 \times 10^{-20}$ | $9.000000 \times 10^{-2}$ | (9,-2) | $1.484577 \times 10^{-16}$ | $5.000000 \times 10^{-4}$ | (5,-4) |
| 9 | $2.609964 \times 10^{-20}$ | $5.000000 \times 10^{-8}$ | (5,-8) | $1.444577 \times 10^{-16}$ | $8.000000 \times 10^{-4}$ | (8,-4) |
| 10 | $1.834337 \times 10^{-20}$ | $7.000000 \times 10^{4}$ | (7,4) | $1.318231 \times 10^{-16}$ | $6.000000 \times 10^{-5}$ | (6,-5) |

*Note:* For CE, probabilities for incorrect classes are small but spread over very large and very small values. For MALL+CE, the residual probability mass is concentrated on nearby values.

**Experiment 3: generalisation to unseen numeric tokens using Triangle Loss.** The model comprises an embedding layer ($153 \times 32$) followed by a two-hidden-layer MLP ($32 \rightarrow 64 \rightarrow 64 \rightarrow 1$) with ReLU and a final Softplus activation to ensure positivity. The 153 tokens (each representing a numeric value) are sorted by their true numeric magnitude; 2/3 are used for training, while the remaining 1/3 are completely unseen (never appear in any training context). The base loss is the MALL distance, which handles the wide numerical range. Additionally, a triangle loss that exploits the known total order of the tokens is added only once every 9 epochs. The loss on train (seen) and test (unseen) tokens is illustrated in Figure 15. Results: adding the triangle loss reduces the test MALL error from approximately 9 to approximately 2.5, bringing it close to the training loss level. This strong generalisation is possible because neural networks learn a continuous function; the triangle loss enforces a well-ordered embedding space, allowing accurate

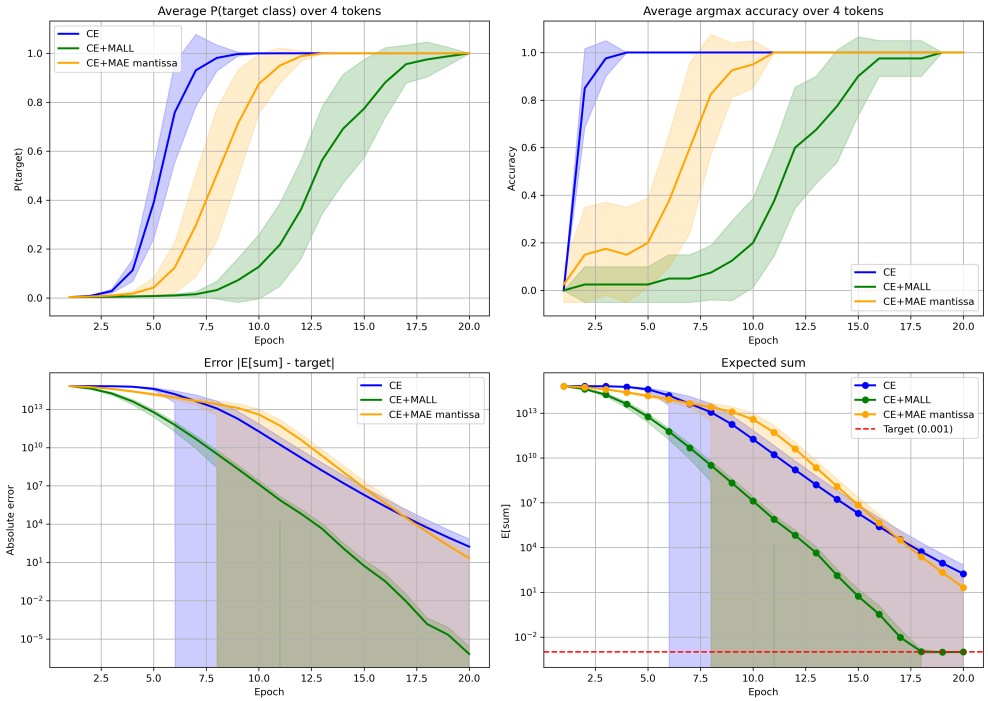

Figure 13: MALL combined with CE, $\lambda = 0.02$

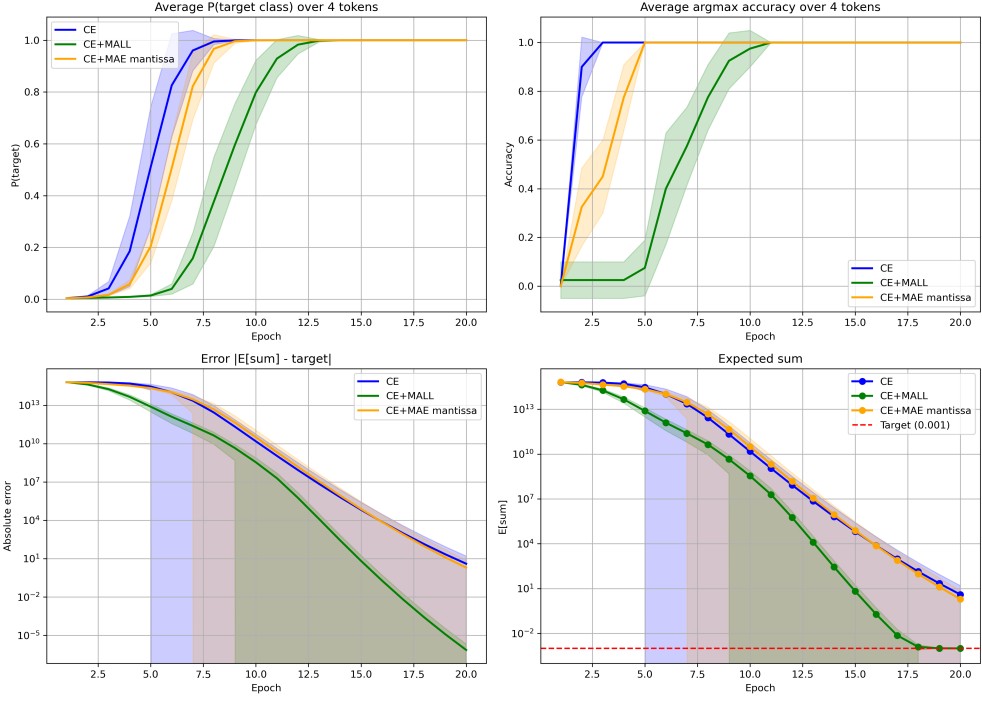

Figure 14: MALL combined with CE, $\lambda = 0.002$

interpolation to unseen tokens. An interesting side effect is that the triangle loss also facilitates learning of the embeddings for the seen tokens.

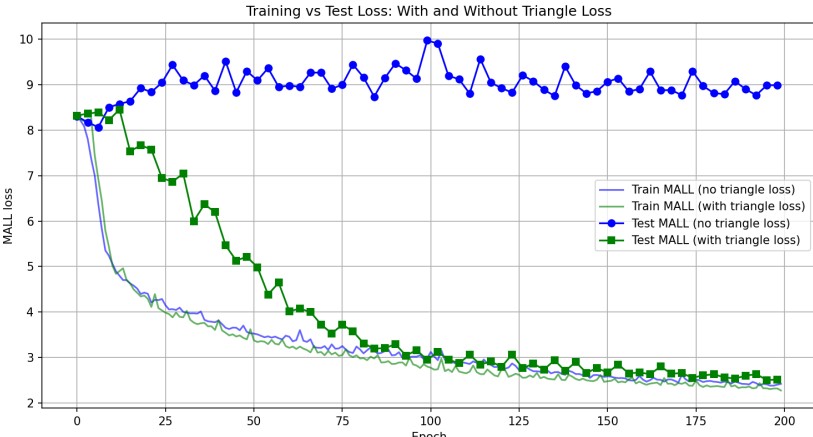

Figure 15: Triangle Loss, test tokens were not seen during training.

