# OpenReview forum: "Number Value Loss in LLMs and N-adic Tokenization"
_TMLR — Decision pending for TMLR_

### Review · Reviewer_WyQw · 2026-05-26

**Summary Of Contributions:**

This paper studies a specific but important weakness of current language models: their difficulty in representing and optimizing numerical values across very small and very large scales. The authors argue that the problem is not only caused by tokenization, but also by the loss landscape used to train models on numerical tokens or decoded numerical values.

The main contribution is the proposed Magnitude-Aware Log Loss, or MALL, which is designed to preserve useful optimization signals across a wide numerical range. The paper motivates this loss through analytical comparisons against MAE/NTL, MSE, and MSLE, and argues that MALL better balances local mantissa precision with global order-of-magnitude correction. A second contribution is the analysis of the “Softmax boundary problem,” where independent token probabilities can produce numerically invalid or far-away “phantom” numbers even when individual tokens appear locally plausible. The paper also presents a no-go style theorem arguing that continuous additive per-token losses cannot remain numerically stable over very large ranges. Finally, the paper proposes Triangle Loss as a geometric embedding regularizer intended to preserve numerical continuity among neighboring numerical tokens, especially for rare tokens in structured numerical tokenizations.

Overall, I found the paper interesting and potentially useful for researchers working on numerical reasoning, tokenization, and loss design for language models. The main strengths are the clear motivation, the focus on an under-addressed optimization issue, the useful analytical comparisons against standard losses, and the attempt to connect tokenization, loss design, and embedding geometry into one coherent framework. The main weaknesses are that the paper is mostly theoretical, lacks empirical validation on trained models or benchmark tasks, and sometimes makes broad claims that would benefit from clearer assumptions and stronger evidence. The presentation also needs some polishing, especially around notation consistency and the distinction between MALL as a metric, loss function, and training objective.

**Additional Comments:**

I enjoyed reading this paper. The core problem is important, and the paper offers a thoughtful perspective on the relationship between tokenization, numerical value representation, and optimization. The proposed MALL objective is intuitive and appears promising from the analytical evidence presented. The discussion of boundary behavior and phantom numerical sequences is also a useful contribution.

My main concern is that the paper currently reads more like a strong theoretical proposal than a complete TMLR submission. The analytical evidence is useful, but the paper would be much stronger with even modest empirical validation. I do not think the authors need to train a very large model to make the paper convincing. A controlled synthetic benchmark or small-scale language-model experiment would already help demonstrate that the proposed loss improves actual learning behavior rather than only improving plotted loss landscapes.

I would encourage the authors to revise the paper by tightening the theoretical assumptions, adding experiments, improving notation consistency, and softening claims that are not empirically demonstrated. With these changes, the paper could become a valuable contribution to the study of numerical reasoning in neural language models.

**Audience:**

Yes

**Audience Explanation:**

Yes. The topic is relevant to TMLR’s audience because numerical reasoning remains a known weakness of language models, and this paper approaches the issue from an optimization and representation perspective rather than only from prompting, data, or benchmark design. Researchers working on tokenization, numerical embeddings, arithmetic reasoning, regression-style objectives for language models, and structured representations would likely find the analysis useful.

The paper is especially interesting because it tries to explain why a structurally better numerical tokenization may still be insufficient if the training objective does not provide stable gradients across scales. The discussion of boundary effects caused by independent token distributions is also relevant beyond the specific N-adic tokenization setting, since similar issues can arise whenever numbers are generated as sequences of discrete tokens.

The current version may be more appealing to readers interested in theory and loss design than to readers looking for empirical improvements on benchmark tasks. Adding experiments would broaden the paper’s appeal substantially.

**Broader Impact Concerns:**

I do not see major negative broader impact concerns requiring rejection. The paper is mainly about improving numerical representation and optimization in language models. Better numerical reasoning could have positive impacts in scientific, financial, engineering, medical, and educational settings where current language models often make unreliable numerical errors.

However, the authors should include a short broader impact statement discussing both benefits and risks. The main risk is that improved numerical fluency may make model outputs appear more trustworthy even when the model is still wrong or poorly calibrated. In high-stakes domains such as finance, healthcare, legal decision-making, or infrastructure planning, better numerical formatting or lower numerical error does not automatically imply reliable reasoning. The paper should explicitly state that improved numerical objectives should be paired with calibration, verification, uncertainty estimation, and domain-specific validation before deployment in high-stakes settings.

The authors should also mention that numerical losses can encode design choices about which errors matter more, such as relative error versus absolute error or magnitude alignment versus local precision. These choices may have application-dependent consequences and should be evaluated carefully.

**Claims And Evidence:**

Yes

**Claims Explanation:**

The paper provides a reasonable amount of analytical evidence to support its main claims. The derivations and visualizations give useful intuition for why standard objectives such as MAE, MSE, and MSLE can become poorly behaved when numerical values span many orders of magnitude. The MALL construction is well motivated, especially the combination of a local relative-error-like term with a separate magnitude penalty. The examples around small values, large values, and boundary transitions are helpful and make the problem concrete.

That said, the evidence is not fully convincing yet because most of the support is analytical or based on manually constructed loss landscapes rather than model-training experiments. The paper claims that MALL can improve numerical learning and that Triangle Loss can help rare numerical tokens, but these claims would be much stronger if supported by experiments on actual neural models. For example, even a small controlled experiment comparing cross-entropy, NTL/MAE-style loss, MSLE, and MALL on numerical prediction tasks would make the contribution much easier to evaluate. Similarly, the Triangle Loss proposal is plausible, but the paper does not yet demonstrate that it improves downstream numerical generalization or rare-token behavior in practice.

I would therefore say that the claims are mostly supported at the level of theoretical motivation and analytical plausibility, but the paper needs additional empirical evidence before the stronger practical claims can be fully accepted.

**Requested Changes:**

The paper would benefit from the following changes before acceptance.
1. The authors should add empirical validation. This is the most important requested change. The paper currently makes several claims about optimization behavior and potential training benefits, but the evidence is mostly theoretical and visual. A small but well-designed experiment would significantly strengthen the paper. For example, the authors could train a simple transformer, sequence model, or classifier/regressor on synthetic numerical prediction tasks and compare cross-entropy alone, MAE/NTL-style loss, MSLE, and MALL. The evaluation could include exact numerical accuracy, relative error, order-of-magnitude accuracy, and performance across small, medium, and large numerical ranges.

2. The paper should clarify the exact scope of MALL. At different points, the paper presents MALL as a metric, a loss function, a drop-in replacement, and a training objective. These uses are related but not identical. The authors should explicitly distinguish when MALL is being used as an evaluation metric, a differentiable expected loss over token probabilities, a scalar regression loss after decoding, or an auxiliary regularizer.

3. The no-go theorem should be presented more carefully. The theorem is interesting, but the assumptions should be stated more explicitly. In particular, the authors should clarify the precise class of tokenizations, loss decompositions, continuity assumptions, and whether the argument applies to training-time token losses, decoded-sequence losses, or both. Some readers may find the current proof too compressed and may question whether the conclusion is as general as stated.

4. The Softmax boundary problem section should be improved with clearer notation and at least one additional example. The 1999/2000 example is helpful, but the section would be stronger if it separated the issue of independent positional uncertainty from the specific NST representation. It would also help to explain when this problem is severe in practice and how often it might occur during training or inference.

5. The Triangle Loss contribution needs either empirical support or a more modest framing. The idea is reasonable, but the paper currently does not show that the proposed embedding correction improves model behavior. If experiments are not added, the authors should describe Triangle Loss as a proposed future direction or structural prior rather than as a demonstrated improvement.

6. The paper needs editing for clarity and polish. Some passages contain grammatical issues, inconsistent terminology, and formatting problems. For example, terms such as TST, NST, N-adic tokenization, triadic tokenization, MALL, DNL, and Triangle Loss should be introduced consistently and summarized in a notation table. Several figures would also benefit from clearer captions, axis labels, and explanation of what exactly is plotted.

7. The related work section should be expanded. The paper cites relevant work such as xVal, numerical token loss, and numeracy probing, but the positioning could be clearer. The authors should explain more directly how MALL differs from existing scale-aware losses, relative-error losses, log-cosh/log losses, Huber-style robust losses, and regression-based number-token methods.

---

> ### Author Response · Authors · 2026-06-13
> **Response to Reviewer and the changes.**
>
> Response to Reviewer and the changes
>
> Dear Reviewer,
>
> Thank you very much for your thoughtful and encouraging feedback. We are genuinely glad that you enjoyed reading the paper. Your detailed comments and constructive suggestions have helped us improve the manuscript significantly.
>
> The revised paper has grown from 16 to 21 pages (including the appendix). Rather than reproducing large blocks of text, we summarise the main changes below with references to the relevant sections. We hope this makes it easier for you to locate the updates.
>
> Below we summarise the main changes we have made in response to your requests.
>
> 1) Empirical validation.
> Following your suggestion, we added three synthetic experiments in the Appendix: order-of-magnitude classification, multi-token sum prediction, and Triangle Loss generalisation. These provide direct empirical support for MALL and Triangle Loss, which was missing in the original submission. The experiments code is also loaded as supplementary files.
>
> 2) Scope of MALL.
> We added a clear table (Section 3) that distinguishes the four different roles of MALL: metric, hardmax regression loss, expected loss over token probabilities, and auxiliary regulariser.
>
> 3) No-Go theorem.
> We expanded the assumptions and clarified the setup (Section 3.4). We now explicitly show how F is computed for different tokenizations (TST and BPE) and added a note on generalisation. The theorem is now stated more precisely.
>
> 4) Softmax boundary problem.
> We improved this section (Section 3.5, example 1999/2000) – explained that the issue occurs at all digit boundaries.
>
> 5) Triangle Loss.
> We provided empirical evidence in Experiment 3 (Appendix) and described the geometric motivation and post‑processing variant in Section 4.
>
> 6) Clarity and polish.
> We added a notation table (Section 2), improved figure captions (especially for b1.png, b2.png, bpe_tst_full.png), and unified terminology (TST, UST, NST).
>
> 7) Related work.
> We expanded the comparison with Huber and log‑cosh in Section 3.3 (last paragraph before the comparative table) and in the Introduction section.
>
> 8) Broader Impact.
> We added a Broader Impact statement after the Conclusion, discussing benefits, potential risks (over‑trust, poor calibration), and recommended precautions.
>
> We believe these changes address all your concerns. Once again, thank you for your valuable time and for your kind words about our work. We hope the revised manuscript is now suitable for publication in TMLR.
>
> Sincerely,
> The Authors

---

> ### Author Response · Authors · 2026-06-20
> **Addition to the request 3)**
>
> To clarify the practical meaning of the No-Go Theorem, we have added Corollary 1 (Paradox of Monotonicity Counter-Intuition) with a strict proof by contradiction and an accompanying Remark in Section 3.2. We demonstrate that any non-continuous additive loss is mathematically forced to exhibit downward jumps to avoid gradient explosion, thereby violating global numerical monotonicity and forcing the architecture to treat larger values as states of lower penalty F(x_high) < F(x_low). Effectively, this means that to represent and understand numbers via additive tokens, LLMs are fundamentally forced to inherently process boundary values under the inversion that 2000 < 1999 for at least some of these critical transitions.

---

### Review · Reviewer_uPAr · 2026-06-04

**Summary Of Contributions:**

The paper presents an analysis of the failure modes of tokenization for numbers in language models, and how it relates to the loss given to these tokens.  The paper proposes a Magnitude-aware Log Loss to preserve the magnitude and the mantissa of the numbers, and provide a high level analysis of the softmax boundary problem.

Strengths:
- The idea of look at the number representations on the language models and their point of failure is interesting.
- The analysis of the loss based on tokens (despite relying on base 10) is interesting, and poses an interesting stepping stone to further study the problem.

Weaknesses:
- The paper proposes the ideas, but does not evaluate nor validate them in any model.
- There are no empirical evaluation of how any model can translate the theoretical bounds and how they behave in practice.
- The softmax boundary problem doesn't consider the practical distribution of numbers in the training data.  It will be interesting to see how the bias in the input distribution affects the theoretical results as well.

**Audience:**

No

**Audience Explanation:**

The paper addresses a niche problem in language models.  The paper fails to contextualize these errors in practice and how prevailing they are in existing work.  The paper references a workshop paper but fails to broaden the motivation in the existing LLM models.

Given this limitation, I consider the TMLR's audience to not be as interested in this paper.

**Claims And Evidence:**

Yes

**Claims Explanation:**

The paper claims to present theoretical results and analysis and delivers on it.  While the scope is limited and the results constrained to theoretical results as pointed by the authors, it delivers on the promises (even if these are limited).

**Requested Changes:**

- The motivation behind better tokenization on LLMs for numbers needs to be improved.  While the introduction mentions that the existing metrics fail, there are no examples or references to particular fail modes in existing methods that are tracked back to the metrics referenced.

- It is not clear from the training description how this metric will be used in practice.  When a LLM is trained, how are the tokens from text are going to be distinguished from the numbers?

  Moreover, depending on the tokenization scheme, to get the full number one must scan until the end of the number and tokenize it afterwards.  What is the overhead of this process in contrast to a traditional tokenizer?

- The setup in Section 3.2 needs to be clarified.  It is not clear how the continuous function f is additive to be separated as proposed.

- In Section 3.2, f in (0,1) is not defined.

- Given the limits of the range [10^-15, 10^18], wouldn't a log of the loss stabilize it?  (Similar logic to your proposed MALL loss and commonly used in probabilistic methods where the values quickly run outside of machine precision.)



Minor comments:
- P3: TST is not defined.
- Figure in Section 3.4 has no caption nor number.

---

> ### Author Response · Authors · 2026-06-18
> **Response to the review and the changes.**
>
> Dear Reviewer,
>
> Thank you for your thoughtful comments.
> Below we address each point in turn, with references to the corresponding changes made in the revised manuscript.
>
> 1) "The motivation behind better tokenization on LLMs for numbers needs to be improved.  While the introduction mentions that the existing metrics fail, there are no examples or references to particular fail modes in existing methods that are tracked back to the metrics referenced."
>
> In this paper, we do not argue that LLMs need a different tokenization; this is out of the scope of our work. Instead, this study investigates the behavior of various loss functions in neural networks from the perspective of value prediction and the underlying relationships between numerical values and tokenizations. The closest practical metric link in LLMs is the NTL paper, which demonstrates improvements in LLM numerical reasoning by adding MAE to the CE loss.
> We should also mention that MALL (Magnitude-Aware Log Loss) is introduced as a robust, scale-invariant loss function designed to provide stable gradient signals for any neural network task dealing with such multi-scale continuous data. LLMs are a prominent application field, but MALL's mathematical utility is general.
> As for the No-Go theorem, it concerns any tokenization of numbers and is a separate result. This theoretical contribution aligns with TMLR's "experimental and/or theoretical studies yielding new insight into the design and behavior of learning in intelligent systems."
>
> Changes:
> In the updated manuscript, we have added three synthetic experiments. Experiment 1 and Experiment 2 illustrate on small networks the difference between MALL and other metrics (including the MAE used in NTL) as both a pure classification and a regression component. We have also added a table (Section 3) that distinguishes the roles of MALL.
>
>
> 2) "...When a LLM is trained, how are the tokens from text are going to be distinguished from the numbers?... What is the overhead of this process in contrast to a traditional tokenizer?"
>
> Distinguishing tokens is a solved problem (e.g., via dedicated token IDs in xVal or NTL), with the only difference being that TST explicitly captures the true numerical magnitude. Whether constructing numbers from BPE or digit-by-digit tokens, the overhead is negligible since the pipeline simply aggregates tokens until the number ends to calculate the loss. If you mean initial input tokenization (like R2L), it is a standard, one-time pre-parsing step. In practice, traditional tokenizers already heavily split standard text words into sub-word tokens anyway, adding no extra penalty for numbers.
>
> 3) "The setup in Section 3.2 needs to be clarified. It is not clear how the continuous function f is additive to be separated as proposed.
> In Section 3.2, f in (0,1) is not defined."
>
> We have added an explanatory block at the beginning of Section 3.2 that clarifies what we mean by an additive function F for the tokenization and provides a formal definition for function f.
>
> 4) "wouldn't a log of the loss stabilize it?"
>
> While log objectives solve magnitude issues, they collapse the mantissa, making the model blind inside the token. As shown in Table 2 (Section 3.3), at macro scales (10¹¹ vs 1.01 × 10¹¹), both MSLE and pure log‑loss (T₂) register 0.000, whereas T₁ provides a robust signal. At micro scales (10⁻⁶ vs 2 × 10⁻⁵), MSLE collapses due to its +1 shift and T₂ fails, while T₁ maintains a strong gradient. This proves that pure log or shifted metrics lack fine‑grained precision, making the local component (T₁) strictly essential.
>
> We have added a discussion of these cases right before the table.
>
>
> 5) "P3: TST is not defined."
> We have added a notation table (Section 2) and unified terminology (TST, UST, NST).
>
> 6) "Figure in Section 3.4 has no caption nor number."
> We have added the caption, number and link from the text.
>
> 7) "Weaknees : ... It will be interesting to see how the bias in the input distribution affects the theoretical results as well."
> It is more a structural question. CE treats each token sequence as a distinct categorical class, meaning it considers 1-999 and 2-000 very far apart (maximum Hamming distance of 2). Consequently, CE distorts numerical reality: it sees 1-0-0-0 and 9-0-0-0 (distance 1) as closer than 1-0-0-0 and 1-0-1-1 (distance 2). This geometric distortion is intrinsic to CE and independent of the training distribution. LLMs optimizing solely on CE lack the mechanism to evaluate full numerical values, whereas MALL explicitly encodes numerical proximity, penalizing the distance to 9000 far more than to 1011 when compared to 1000.
>
> If the reviewer considers this additional discussion on CE geometric distortions a valuable insight, we would be glad to formally integrate it into Section 3.4.
>
> Sincerely,
> The Authors

---

> > ### Author Response · Authors · 2026-06-20
> > **Addressing the interest and practical relevance of the findings**
> >
> > To further increase the interest of TMLR readers, we have added Corollary 1 (Paradox of Monotonicity Counter-Intuition) to the No-Go Theorem with a strict proof. We prove that for some boundary transitions, LLMs are fundamentally forced to exhibit F(x_high) < F(x_low), meaning that in terms of loss LLMs would think that 2000<1999 or similar points.

---

> > > ### Comment · Reviewer_uPAr · 2026-06-24
> > >
> > > I thank the authors for the reply.  While their answers addressed my comments, I have more concerns about the proposal and share some of the concerns from reviewer SMe5.
> > >
> > > The claims are still covered, but given the explanations from the authors, I think that the audience for TMLR is far away from the results presented despite the authors efforts.

---

> > > > ### Author Response · Authors · 2026-06-25
> > > > **Official Comment by the Authors**
> > > >
> > > > We appreciate all feedback. However, regarding the question of audience interest:
> > > >
> > > > 1) The criterion is "at least some individuals". Reviewer WyQw explicitly answered "Yes" and gave a detailed justification. The condition is satisfied.
> > > >
> > > > 2) Reviewer SMe5 openly stated the topic is "beyond my direct domain and my expertise".
> > > >
> > > > 3) Furthermore, our No‑Go theorem reveals a fundamental paradox: tokenized neural networks are mathematically forced to treat 2000 < 1999 in terms of loss (or at other certain boundaries). This striking result addresses a core limitation in every LLM that handles numbers split into tokens and, we believe, will interest theorists and will be clear and easy to understand for anyone interested in the workings of LLMs.
> > > >
> > > > We agree that the paper is mathematically complicated, but it reveals fundamental problems and goes along with TMLR's goal to publish "experimental and/or theoretical studies yielding new insight into the design and behavior of learning in intelligent systems."
> > > >
> > > > Sincerely,
> > > > The Authors

---

### Review · Reviewer_SMe5 · 2026-06-17

**Summary Of Contributions:**

This paper is beyond my direct domain and my expertise. I try to understand it and appreciate the theoretical explanation provided by the authors. Here are the contributions I can acknolwedge after reading the paper carefully:
1. The authors introduce Magnitude-Aware Log Loss (MALL).
2. The authors analyze the loss landscapes of different approaches, including two tokenization variants (BPE, TST) and two loss functions (MALL, NTL) and identify the limitations of NTL loss based on the loss landscape curves.
3. The authors conduct some theoretical analysis to explain the observed loss landscapes.

**Audience:**

No

**Audience Explanation:**

1. Mathematical understanding and reasoning is an important capability of AI models and tokenization is an important aspect for achieving this capability. While this research topic would interst many TMLR audience, the paper provides no review of related work and it is hard for TMLR audience to understand the specific contribution of this paper.
2. From my perspective, this paper is very hard to understand. Currently, it's hard to know whether the major contribution of the paper is a new loss format or the theoretical contribution, or something else.

**Broader Impact Concerns:**

None.

**Claims And Evidence:**

No

**Claims Explanation:**

To be honest, I am unsure what's the main claim of this paper. The paper is titled as "Number Value Loss in LLMs and N-adic Tokenization", so I assume the paper is about analyzing different number value loss functions in LLMs and N-adic tokenization.

However, in the abstract, the paper argues that it addresses the root causes of optimization failure in numerical representations. If this is the main argument, I think the paper needs to conduct emperical experiments on math-related tasks and compare the proposed method with other available methods in this area.

> Through mathematical proofs and gradient field visualizations, we demonstrate that our framework addresses the fundamental limitations of current numerical objectives, providing a robust foundation for coherent numerical intelligence in neural architectures.

I do not think gradient field visualizations are sufficient as the emperical evidence here. Also, for the mathematical proofs, it is unclear whether modern LLMs hold those assumptions required by the proof.

**Requested Changes:**

1. Add a related work section to help audience understand how this work contributes to this research area.
2. Refactor the paper to make it clear on the prime contribution of this paper.

---

> ### Author Response · Authors · 2026-06-19
> **Response to the review and the changes.**
>
> Dear Reviewer,
>
> We appreciate your time and the effort you spent reviewing our paper. Since our core theoretical claims and architectural focus span multiple interconnected areas, we welcome this opportunity to clarify the manuscript's structure and contents.
>
> 1. Add a related work section to help audience understand how this work contributes to this research area.
>
> We would like to clarify that a comprehensive review of related work is fully integrated into Section 1 (Introduction), as this structure is essential for establishing the direct theoretical motivation behind our architectural claims.
> The introduction initially provides 12 distinct citations to relevant, foundational, and state-of-the-art methodologies in the field. These include standard BPE tokenization (Sennrich et al., 2015), Llama 3 (Dubey et al., 2024), tokenization failure modes (Zhang et al., 2025), xVal (Izacard et al., 2023), and NTL (Zausinger et al., 2025). Furthermore, since this paper focuses on the design of loss functions, we extensively compare MALL against core objectives used for multi-scale numerical data: classical robust losses like Huber (Huber, 1964), logarithmic variants such as MSLE, recent specialized surveys (Li et al., 2025), as well as standard MSE and MAE. We have now also added log-cosh loss  (Chen et al., 2019) for comparison.
>
> 2. Refactor the paper to make it clear on the prime contribution of this paper.
>
> This work investigates the behavior of various loss functions in neural networks from the perspective of value prediction and the underlying relationships between numerical values. We address multiple distinct facets of this problem. As stated in the abstract, introduction, and conclusion, the main outcome of this paper is not a single isolated contribution, but rather a set of multiple, interconnected yet independent results. Specifically, as documented throughout the manuscript, these independent findings include:
> - the introduction of the MALL metric alongside its comprehensive evaluation against standard baselines for this task;
> - the analysis of the Softmax boundary problem together with the structural No-Go theorem for loss functions in tokenized neural networks;
> - and the Triangle Loss regularizer designed for application at the embedding manifold level.
>
> 3. Regarding Empirical Evidence and "Math-Related Tasks":
> "I do not think gradient field visualizations are sufficient as the empirical evidence here. Also, for the mathematical proofs, it is unclear whether modern LLMs hold those assumptions required by the proof."
>
> Analytically investigating the gradient field and providing theoretical proofs—such as our Lemma (Properties of MALL)—is the only definitive way to objectively prove a loss function's viability and stability across an entire continuous spectrum. While discrete neural network experiments can show practical applicability in specific settings, they cannot mathematically guarantee a global theoretical claim across a 33-order magnitude range. Furthermore, we have examined the most widely used numerical benchmarks, such as NumericBench. While these benchmarks are excellent for testing standard arithmetic reasoning, their evaluation data is inherently structured around local, fixed-point distributions (e.g., configurations such as 2098.7986965636 or 7226). Consequently, they do not simulate the extreme 33-order magnitude shifts that our framework is specifically designed to address, making downstream testing on them less applicable for evaluating global theoretical stability.
> However, to demonstrate that our theoretical findings hold in practice, the updated manuscript now includes three synthetic experiments in the Appendix. These controlled experiments confirm that the mathematical advantages of MALL directly translate into robust optimization behavior in neural architectures. Additionally, regarding the assumptions of the No-Go theorem, we have added a dedicated discussion block to Section 3.2 explaining that the theorem is universal across different tokenization schemes, explicitly including standard BPE.
>
> 4. "The authors analyze the loss landscapes of different approaches, including two tokenization variants (BPE, TST) and two loss functions (MALL, NTL) and identify the limitations of NTL loss based on the loss landscape curves."
>
> We would like to clarify that the main comparison of MALL is provided in Section 3.3 "Gradient Field Analysis: MALL vs. MSLE and MAE". Table 1 of the initial paper also contains data on MSE. We now have also added some discussion on Huber and log-cosh losses there.
>
>
> Sincerely, The Authors